# Atropostatin: Design and Total Synthesis of an Atropisomeric Lactone–Atorvastatin Prodrug

**DOI:** 10.3390/molecules28073176

**Published:** 2023-04-03

**Authors:** Daniel Pecorari, Andrea Mazzanti, Michele Mancinelli

**Affiliations:** Department of Industrial Chemistry “Toso Montanari”, University of Bologna, Viale del Risorgimento 4, 40136 Bologna, Italy; daniel.pecorari2@unibo.it (D.P.); andrea.mazzanti@unibo.it (A.M.)

**Keywords:** atropisomer, atorvastatin, atropostatin, dynamic NMR, docking calculations, ECD, DFT, TD-DFT calculations

## Abstract

Atorvastatins play an important role in the inhibition of HMG-CoA reductase, an enzyme present in the liver that takes part in the biosynthesis of cholesterol. In this article, we report the total synthesis of a lactone–atorvastatin prodrug with additional atropisomeric features. Conformational and experimental studies of model compounds were designed to test the stability of the chiral axis. Docking calculations were performed to evaluate the constant inhibition of a library of atorvastatins. Full synthesis of the best candidate was achieved and thermally stable atropisomeric lactone–atorvastatin was obtained. The absolute configuration of the chiral axis of the atropisomers was assigned by means of chiroptical ECD spectroscopy coupled with TD-DFT calculations.

## 1. Introduction

The biological activity of drugs relies on the ability to interact with enzymes and receptors, mainly changing the activity of a selected active site. In this contest, chirality plays a fundamental role for the interactions in a biological system, increasing the activity and selectivity of the drug. Chiral organic molecules interact differently with the active sites of enzymes in different chemical–physical ways in order to favor specific spatial conformations [1].

Almost the totality of chiral drugs bears stereogenic carbons as their source of chirality, while axial chirality in pharmaceutical research has usually been considered as an issue, instead of an opportunity. The troubles arise from their difficult synthesis and kinetic axial control in a biological environment [2]. However, atropisomers have been found in naturally occurring compounds and their increase in activity and selectivity towards the respective enzymes has undoubtedly been evident [3,4,5,6,7].

Recently, many atropisomeric drugs or atropisomeric modifications of known drugs have been reported in the literature, which is a sign of a new field emerging within drug development research, and many atropselective syntheses have been presented [8,9,10,11,12,13,14].

Atorvastatin is well known for its inhibition of the enzyme 3-hydroxy-3-methylglutaryl-coenzyme A reductase (HMGR), fundamental for the synthesis of endogenous cholesterol. Atorvastatin is effective at reducing levels of total cholesterol and low-density lipoprotein (LDL) cholesterol, and is mainly prescribed for patients with dyslipidemia and cardiovascular disease [15]. Besides its considerable number of benefits, the drug also demonstrates serious side effects, such as muscle breakdown, kidney damage or liver problems, or less serious side effects, such as nausea, headaches and nosebleeds [16,17,18]. Consequently, the improvement of statin effectiveness is an ambitious target to lower the active dosage and hence to reduce side effects.

The chemical structure of atorvastatin combines an enantiopure dihydroxyheptanoate-chain bonded to the nitrogen of a decorated pyrrole scaffold (Figure 1). The chiral portion has been found to be essential for the inhibition of the HMGR [19]. Positions 2 and 5 of pyrrole are occupied by *p*-fluorophenyl and isopropyl groups, respectively; position 4 is occupied by a phenyl amide and position 3 by a phenyl group. The substitutions in positions 2 and 5 are dictated by the geometric shape of the active site of the enzyme, while the substitutions in positions 3 and 4 raised the efficacy of the drug as shown in SAR studies.

Herein, we have reported a library of atorvastatins with structural modifications at position 3 of pyrrole to develop an additional chiral axis with a racemization barrier bigger than 30 kcal/mol (Class III atropisomers in the Laplante terminology [4,5]), yielding consequently diastereomeric drugs with potentially different pharmacological effects. DFT calculations and kinetic experiments were used to evaluate the rotational racemization barrier in atorvastatin-model compounds lacking the chiral chain (Section 2.1 and Section 2.3). Then, preliminary docking simulations were performed to assess the biological activity vs. standard atorvastatin (Section 2.4). Finally, a new synthetic pathway for the total synthesis of the best candidate was also presented, obtaining diastereoisomeric lactone–atorvastatin pro-drugs (Section 3).

## 2. Results and Discussion

### 2.1. DFT Calculations

The stability of the chiral axis, at human body temperature (37 °C), is of fundamental importance in order to consider the atropisomer as a single-entity drug. Therefore, we first evaluated the energy barriers for racemization of the chiral axis using an atorvastatin model library (Figure 1).

The opportunity to simplify the structure of atorvastatin resulted in faster DFT calculation and easier, more cost-effective, compound synthesis (see Section 2.2). The modifications must not alter the steric environment of the chiral axis, thus obtaining highly representative results of the complete atropostatins (Figure 1).

In the atorvastatin model structures, the *p*-fluorophenyl at position 2 of pyrrole was substituted by a phenyl group and the dihydroxyheptanoate chain was replaced by a methyl group. The stereogenic axis at position 3 of the pyrrole was realized by using an asymmetric aryl ring, which was forced out of the planar pyrrole plane, with the skew angle being larger the bigger was the steric hindrance in the ortho positions. The asymmetry of aryl and the perpendicularity with pyrrole plane generates ***P*** and ***M*** atropisomeric conformations [20,21].

***P/M*** interconversion (i.e., racemization of the chiral axis) was investigated by calculating the ground and transition states for compounds **1a–1h** with density functional theory (DFT) using B3LYP-D3/6–31G(d) with empirical dispersion [22].

In Figure 1, we report the atorvastatin model **1a** having an *o*-tolyl group at position 3 of pyrrole. All the pyrrole substituents in ***GS-M*** generate a helical geared system where the torsional motions of rings are strongly correlated (Figure 1). Therefore, the rotation of the *o*-tolyl group drives the movement of the other substituents in the same direction.

In the lowest energy ***GS-M***, the isopropyl CH proton is close to the NMe-pyrrole, whereas the carbonyl of the amide is directed toward the isopropyl substituent. The evaluation of the energy rotational barrier of 3-aryl groups was obtained by modelling two plausible TS starting from the ***GS-M***. The expected TS must have the aryl coplanar with the pyrrole ring in order to convert the chiral axis from ***GS-M*** to ***GS-P***. The TS0 regards the *o*-methyl substituent close to the carbamate, while in TS180 it is on the phenyl side at position 2. In the TS geometries, the carbamate and the phenyl are moved to be perpendicular to the pyrrole to allow space at the planar aryl. The ***GS-M*** and ***GS-P*** interconversion occurs by the lowest energy path between TS-0 and TS-180. With these considerations, we optimized the GS geometries and the rotational energy barriers of a library of pyrroles with different aryl steric hindrance at position 3. The results have been summarized in Table 1 (see Section 2.3). The pyrroles **1a** and **1b** have a rotational energy barrier too low to be configurationally stable at +37 °C (23.4 kcal/mol and 24.5 kcal/mol, respectively), while **1c–f** have an estimated rotational energy barrier higher than 31 kcal/mol, suitable for a biological study.

### 2.2. Synthesis of Atorvastatin Models

The experimental rotational energy barriers of atorvastatin models were determined after planning a feasible synthetic route with these bulky aryl substituents. A good survey of examples of atorvastatin synthesis is reported in the literature [23,24,25], but the aryl steric hindrance at position 3 poses additional issues, lowering the reaction yields. For this reason, we focused our attention on a zirconium-catalyzed method for pyrrole cascade cyclization [26], with appropriate modifications for our purposes. The proposed pathway to cyclization requires the use of an enamine (**2**) and a nitrovinyl–aryl compound (**3**, Figure 2). The enamine **2** was synthesized with high purity starting from commercially available 4-Methyl-3-oxo-N-phenylpentanamide and MeNH_2_ in refluxing ethanol, without needing further purification (Appendix A, I). Additionally, the nitroalkenes 3 were prepared with the classic Henry reaction with quantitative yields (Appendix A, II).

Using enamine **2** and nitrovinyl-aryl **3** with zirconocene catalyst, the pyrrole **4** was synthesized with good yields (up to 67%, Figure 2). Successively, compound **5** was obtained by bromination at position 2 of pyrrole with NBS (quantitative yield). Eventually, a Suzuki coupling reaction yielded the final model products **1a–1f** and **1h**. With this strategy, highly hindered pyrroles were synthesized with very good total yields (up to 97%; see experimental section). Nevertheless, compound **1g** with 2-naphthol aryl substituent could not be synthesized through this strategy. In fact, the cyclization step (first reaction in Figure 2) yielded a large number of by-products, probably due to the presence of the free hydroxyl group on the naphthyl ring. This drawback was promptly solved by preparing compound **1e** (2-methoxy-1-naphthyl) and then deprotecting the methoxy group with BBr_3_ (Appendix A in Appendix A).

### 2.3. Kinetic Studies of Rotational Energy Barriers

With the synthesized atorvastatin model pyrroles **1a–1h** in hand, we investigated the rotational energy barriers of the ***M*** atropisomeric interconversion with kinetic experiments [21,27,28,29]. For the pyrrole **1a**, variable temperature NMR (D-NMR) experiments monitoring the multiplicity of the isopropyl group allowed us to determine a racemization barrier of 21.2 kcal/mol, fully consistent with the DFT calculated data (Appendix A and Table 1) [30].

Although high NMR temperatures were used (>100 °C), no dynamic process was observable in the compound **1b**, highlighting an energy barrier above 21 kcal/mol. On the other hand, the CSP-HPLC chromatogram taken at 25 °C showed a low plateau between the two atropisomer peaks, thus implying that racemization occurred within the column. Thus, chromatograms were recorded at different temperatures (dynamic HPLC) to evaluate the energy barrier and their simulation provided an energy barrier of 21.7 kcal/mol (Appendix A).

As the computed rotational energy barriers of **1e–1h** were very high (31.1 to 37.7 kcal/mol, Table 1), the respective atropisomers were resolved through CSP-HPLC and the racemization barrier was determined by kinetic analysis (Appendix A). A sample of the first eluted atropisomer was heated at constant temperature in 1,1,2,2-tetrachloroethane, and aliquots of the solution were collected at different times and analyzed by CSP-HPLC at 25 °C to measure the atropisomeric excess. For compounds **1c** and **1d**, no racemization occurred at +146 °C after 24 h and the energy barriers were extrapolated as higher than 36 kcal/mol. All experimental and computational data are reported in Table 1, confirming the reliability of our computational results.

### 2.4. Docking of Modified Atorvastatins

The final goal of this research is a feasible synthesis of an atropostatin **II** (Figure 1, right) with a chiral axis.

The most promising candidate within the library of atropostatin **II** was selected using docking simulations using AutoGrid 4.0 and AutoDock4.2 software packages (see Appendix A) [31,32]. Docking simulations were performed to characterize the binding pose of our candidate atropostatin **II** inhibitor of 1HWK complex of the catalytic portion of human HMG-CoA reductase.

We employed 1HWK crystallographic data downloaded from the RCSB Protein Data Bank [33] and we tested the known atorvastatin I (Figure 1, left).

The resulting interactions of the top-scoring model were mostly related to the chiral alkyl chain (Figure 2, left image) and fully consistent with the literature data [33].

The hydroxyl groups of the chiral alkyl chain interacted with the aminoacidic residues located in the cis loop (as Ser684, Arg590, Asp690, Lys691 and Lys692).

The binding mode and affinities of the evaluated compounds within the catalytic sites of the enzyme were calculated for all ***M*** and ***P*** atropisomers of the atropostatins **II** (Figure 1, right and Figure 2, right image).

We analyzed the location and orientation of the candidate compounds and their interactions with the binding sites of the 1HWK crystal structure, and we found the same disposition of known atorvastatin I. The predicted binding free energy (which includes torsional free energy and intermolecular energy) was used as the criterion for ranking. The conformation with the lowest binding free energy value was considered to be the best docking result (Table 2). Ligand efficacy and virtual inhibition constants (k_i_) at nanomolar concentration were calculated.

All compounds were docked and fit satisfactory well into the active sites of 1HWK.

Although the activity of the ***M*** and ***P*** atropisomers of each atropostatin **II** is indeed different, the preference of one stereoisomer over the other was not predictable in most cases. Nevertheless, we found that ***M*-IIg** atropostatin was the most active atropisomer of the library, with a k_i_ of 0.58 nM and binding energy of −12.6 kcal/mol. The k_i_ was much lower than **I** (k_i_ = 4.5 nM) and it was nearly four-fold more active than respective ***P*-IIg** diastereoisomer (k_i_ = 3.91 nM), highlighting an effective difference in activity between the two atropostatin **IIg**. We assume that the particular conformation created by the hydrogen bond between the hydroxyl and the amidic carbonyl create a hydrophobic surface more apt for the active site than the respective atropisomer ***P*-IIg** and atorvastatin **I** itself. Therefore, atropostatin ***M*-IIg** was chosen to attempt a full synthesis.

### 2.5. Full Synthesis of **M-*IIg*** Atropostatin as Lactone Prodrug

As it is known that atorvastatins can be employed in many chemical forms, we decided to prepare the lactone–atropostatine because of its simpler purification and characterization. The lactone form of atorvastatin would be absorbed from the gastrointestinal tract producing the active drug in the liver, thus acting as a prodrug [34].

Firstly, we used the previous synthesis used for the atorvastatin model **1g** with few modifications to obtain atropostatin **1g’** (Figure 3).

For the preparation of the enamine **2′**, bearing the enantiopure chain, we used the commercially available amine *t*-butyl-[6-(2-aminoethyl)-2,2-dimethyl-[1,3]dioxan-4-yl]-acetate instead of methylamine (Appendix A). The reaction was performed in EtOH with 1% mol of Yb(OTf)_3_ as a catalyst at +40 °C [35]. The resulting sticky oil enamine **2′** can be used without further purification, or it can be crystallized from ACN, with a 65% yield. The nitroalkene **3g**, as previously shown, cannot be used; therefore, we used the 2-methoxy-naphthyl nitroalkene **3e**. Using a 1.2 eq excess, the yield of the cyclization step increased to 63%, yielding the pyrrole **4e’** as a couple of stereolable diastereoisomeric conformations, owing to the lack of substituents at position 2 of pyrrole. While the bromination at position 2 occurred with quantitative yield obtaining **5e’**, the following Suzuki reaction to **1e’** appeared to be challenging. In the same reaction conditions of Figure 2, the use of *p*-fluorophenyl boronic acid yielded, after 12 h reaction, significant amounts of de-brominated pyrrole **4e’.** Moreover, purification of pyrrole **4e’** from **1e’** using chromatography on silica showed different results. However, using a semi-preparative CSP-HPLC (Chiralpak AD-H column), it was possible to separate **4e’** from **1e’** and obtain the two atropisomers **1e’** with 22% yields each (Appendix A). However, pyrrole **4e’** could be recycled to prepare **5e’**. Unfortunately, the deprotection step from **1e’** to prepare **1g’** with BBr_3_ failed, resulting in a complex mixture, formed perhaps by the cleavage of the amide group.

Therefore, we switched to a more suitable protecting group, such as the benzyl group, and we were able to obtain the synthesis of atropostastatin **1g’** with few modifications (Figure 3). We noted that an increase in zirconocene catalyst loading affected the yields of the first step in this reaction (53%, **4h’**), along with an increase of temperature. Additionally, a decrease of the temperature in the second step leads to a more selective bromination at position 2 of the pyrrole ring (91%, **5h’**). Successively, the Suzuki reaction allowed us to obtain the pyrrole **1h’**. Additionally, in this case, a non-negligible amount of de-brominated reagent **4h’** was found at the end of the reaction. With a semi-preparative CSP-HPLC separation, it was possible to obtain **1h’** as separated pure diastereoisomers (33% yield of each diastereoisomer after CSP-HPLC separation, Appendix A).

The last step in the synthesis of the prodrug lactone-atropostatin **7** was the deprotection of hydroxyls and *t*-butyl ester (Figure 4).

The mixture of **4h’** and **1h’** was reacted with aqueous HCl (5 N) in THF at reflux for two hours to obtain lactone atropostatin **6h** and **7h** (yield 63%). The solution of lactones was extracted with EtOAc and an easy separation of the two lactones (**6h** and **7h**) was performed in silica gel column. Afterwards, the lactone atropisomers ***M*-7h** and ***P*-7h** were separated by CSP-HPLC, with Chiralpack AD-H column (*n*-Hexane/*i*-PrOH 80:20, flow 20 mL/min, Figure 3). Lastly, the cleavage of the benzyl for each atropisomer was carried out using Pd(OH)_2_/C as a catalyst and Et_3_SiH (10 eq) as the H_2_ source (***M*-7g** and ***P*-7g**, yields 97%). The assignment of the absolute configuration is reported in the next chapter.

### 2.6. Absolute Configuration of **M-*7g*** and **P-*7g*** Prodrug

Despite many attempts, good crystals for compounds **7h** and **7g** could not be obtained; therefore, the anomalous dispersion X-ray diffraction could not be used to assign the absolute configuration (AC) of compounds. Moreover, despite the presence of known chirality along the chain (lactone is *R*,*R*), any NMR NOE experiment could not help to deduce the AC owing to the long distance between the chiral axis and the protons in the chiral chain.

More conveniently, the chiroptical properties and, in particular, the electronic circular dichroism spectrum (ECD) in combination with time-dependent density functional theory (TD-DFT) was used to discriminate the ***M*** or ***P*** chiral axis of the desired compounds [36,37,38,39,40]. The experimental ECD spectra of compounds ***M*-7g** and ***P*-7g** show very similar bands, with the expected opposite trend. For **7g**, obtained from the first eluted **7h**, the typical absorption of the 1-naphthyl chromophore was found as a strong negative CD band at 255 nm. The ECD spectrum also shows 2 other weak positive bands at 230 nm and 285 nm owing to the dipoles of benzyl, phenyl and pyrrole chromophores.

Although the two atropostatin lactone **7g** are diastereoisomers, the final ECD spectrum is not appreciably influenced by the chiral chain, which is not a strong chromophore. For a faster calculation, the ECD spectrum was therefore calculated for the ***M*-7g** model and compared with the experimental diastereoisomer spectra of **7g** (Figure 3).

The computed ECD spectrum of each atropisomer involved the simulation of all the populated GS, followed by their weighted sum by Boltzmann population (Figure 3). For the ***M*-7g** model, four GSs were found depending on the helical disposition of the pyrrole substituents (see Appendix A in Appendix A). Simulations were performed with four different functions (for data redundancy) and the 6-311++G(2d,p) basis set including the acetonitrile as solvent (IEF-PCM approach) [41,42].

In Figure 3, the calculated spectrum for the ***M*-7g** model shows a good overlap with the experimental spectrum of the first atropostatin lactone **7g**. Therefore, the ***M*** configuration at the chiral axis was assigned to the first **7g** diastereoisomer, derived from the first eluted CSP-HPLC **7h** (red). Consequently, the ***P*** AC was assigned to the second diastereoisomer **7g**, obtained from the second eluted **7h** (green).

## 3. Materials and Methods

### 3.1. Chemical Synthesis

Analytical-grade solvents and commercially available reagents were used as received. The commercially available reagents used were: 4-methyl-3-oxopentanoate; *t*-butyl 2-((4*R*,6*R*)-6-(2-aminoethyl)-2,2-dimethyl-1,3-dioxan-4-yl)acetate; sodium bicarbonate; hydrochloric acid; sodium chloride; methylamine (33% w in ethanol); 2-methylbenzaldehyde; 2,3-dimethyl-1-naphthaldehyde; 2-methyl-1-naphthaldehyde; 1-naphthaldeyde; 2-methoxy-1-naphthaldehyde; 2,3-dimethoxy-1-naphthaldehyde; 2-(Benzyloxy)-1-naphthaldehyde; ammonium acetate; nitromethane; sodium acetate; bromine; phenylboronic acid; *p*-fluorophenylboronic acid; potassium carbonate; tetrakis(triphenylphosphine)palladium(0); *N*-bromosuccinimide, zirconocene dichloride; and boron tribromide (1.0 M in dichloromethane).

The following solvents were used: toluene; hexane; ethanol; petroleum ether; dichloromethane; chloroform; CPME; ethyl acetate; demineralized water; and THF anhydrous. Deuterated solvents for NMR spectra are commercially available.

The following stationary phases were employed for the chromatography: silica gel 60 Å F254 (Merck) for TLC and silica gel 60 Å (230−400 mesh), Sigma-Aldrich, Merck Life Science S.r.l. Milano, Italy, for atmospheric pressure chromatography. The glassware used in these reactions was placed in an oven at 70 °C for at least 3 h immediately before use.

A Waters 600 HPLC pump and a Waters 2487 UV detector with a wavelength set at 254 nm were used to purify the products. The columns used are described in the products characterizations.

NMR spectra were recorded using a spectrometer operating at a field of 14.4 T (600 MHz for ^1^H and 151 MHz for ^13^C). ^19^F spectra were recorded at 9.4 T (376 MHz). Chemical shifts are given in parts per million relative to the internal standard tetramethylsilane (^1^H and ^13^C) or relative to the residual peak of the solvents. The ^19^F spectra chemical shifts were relative to the external standard CFCl_3_. The 151 MHz ^13^C spectra were acquired under proton decoupling conditions with a 36,000 Hz spectral width, 5.5 μs (60° tip angle) pulse width, 1 s acquisition time and a 5 s delay time. The ^13^C signals were assigned by distortionless enhancement by polarization transfer spectra (DEPT 1 and 1.5).

HRMS spectra were recorded on a Waters TOF Premier spectrometer.

### 3.2. ECD

The ECD spectra of compounds **7g** were acquired in the 190–400 nm region using a JASCO J-810 spectropolarimeter in far-UV HPLC-grade acetonitrile solution. Concentration was about 1 × 10^−4^ M, tuned by dilution in order to obtain a maximum absorbance between 0.8 and 1 with a cell path of 0.2 cm. The spectra were obtained by an average of 6 scans at 50 nm∙min^−1^ scan rate.

### 3.3. Kinetic Experiments

The racemization rate measurements were performed as follows: an aliquot of a pure atropisomer of **1b** was dissolved in 1 mL of C_2_D_2_Cl_4_ in a two-neck balloon, one neck with a septum for sampling and one neck for inserting a thermometer. The balloon was kept at 50 °C. C_2_D_2_Cl_4_ was chosen because of its high boiling point and good vapor pressure, which allow it to easily evaporate. Small aliquots were taken at different times, the solvent was evaporated and the sample was analyzed by enantioselective HPLC, which allowed the determination of the enantiomeric ratio at different reaction times. A first-order kinetic equation was then used to derive the rate constant for racemization and, hence, the activation barrier using the Eyring equation.

### 3.4. 5-Methyl-3-(methylamino)-N-phenylhex-2-enamide *(**2**)*

To a solution of 4-methyl-3-oxo-*N*-phenylpentanamid (3.72 g, 18.2 mmol) in ethanol (0.5 M, 7.49 mL), methylamine (33% w solution in EtOH, 17.27 mL) was added. The resulting mixture was refluxed for 7 h and concentrated in vacuo, obtaining a yellow oil used without further purification (3.97 g).

^1^H NMR (600 MHz, CD_3_CN, 1.96 ppm, 25 °C) δ 1.14 (d, *J* = 6.97p, 6H), 2.77 (sept. *J* = 6.97 Hz, 1H), 2.92 (d, *J* = 5.35 Hz, 3H), 4.60 (s, 1H), 6.96 (t, *J* = 7.39 Hz, 1H), 7.24–7.28 (m, 2H), 7.52 (d, *J* = 8.54 Hz, 2H), 6.67 (s broad, 1H), 9.21 (s broad, 1H);

^13^C{^1^H} NMR (151 MHz, CD_3_CN, s 118.26 ppm, 25 °C) δ 21.6 (2 CH_3_), 28.9 (CH), 29.1 (CH_3_), 81.2 (CH), 119.6 (2 CH), 122.8 (CH), 129.6 (CH), 141.5 (Cq), 170.8 (Cq), 171.7 (Cq).

HRMS (ESI-QTOF). Calculated for 219.1497 C_13_H_19_N_2_O [M + H]^+^; found 219.1495.

### 3.5. General Procedure for the Synthesis of β-nitrovinyl Aryls *(**3**)*

A stirred mixture of the corresponding aldehyde (3 mmol, 1 eq), ammonium acetate (0.9 mmol, 0.3 eq) and nitromethane (9 mL, 55 eq) was refluxed for 2–12 h and monitored by TLC. Afterward, the mixture was treated with a saturated aqueous solution of NaHCO_3_ and three-fold extracted with CH_2_Cl_2_. The combined organic phases were dried over anhydrous Na_2_SO_4_ and the nitromethane was distilled under reduced pressure. No further purification was needed.

#### 3.5.1. (*E*)-1-methyl-2-(2-nitrovinyl)benzene (**3a**)

Following the general procedure (2 h reaction time). Obtained as a red oil. Total of 460.2 mg, yield = 94%.

^1^H NMR (600 MHz, CDCl_3_, 7.26 ppm, TMS, 25 °C) *δ* 8.31 (d, *J* = 13.6 Hz, 1H), 7.56–7.49 (m, 2H), 7.40 (td, *J* = 7.5, 1.3 Hz, 1H), 7.32–7.23 (m, 2H), 2.49 (s, 3H).

^13^C{^1^H} NMR (151 MHz, CDCl_3_ 77.16 ppm, TMS, 25 °C) *δ* 139.3 (Cq), 137.7 (CH), 136.9 (CH), 132.0 (CH), 131.5 (CH), 129.0 (Cq), 127.5 (CH), 126.9 (CH), 20.0 (CH_3_).

HRMS (ESI-QTOF). Calculated for 164.0712 C_9_H_10_NO_2_ [M + H]^+^; found, 164.0709.

#### 3.5.2. (*E*)-1-(2-nitrovinyl)naphthalene (**3b**)

Following the general procedure (8 h reaction time). Obtained as a yellow solid. Total of 585.7 mg, yield = 98%.

^1^H NMR (600 MHz, CDCl_3_, 7.26 ppm, 25 °C) *δ* 8.84 (d, *J* = 13.4 Hz, 1H), 8.14 (d, *J* = 8.4 Hz, 1H), 8.01 (d, *J* = 8.2 Hz, 1H), 7.93 (dd, *J* = 8.1, 1.4 Hz, 1H), 7.76 (d, *J* = 7.2 Hz, 1H), 7.68–7.62 (m, 2H), 7.60 (ddd, *J* = 8.1, 6.8, 1.2 Hz, 1H), 7.53 (t, *J* = 7.7 Hz, 1H).

^13^C{^1^H} NMR (151 MHz, CDCl_3_, 77.16 ppm, 25 °C) *δ* 138.7 (CH), 136.3 (CH), 133.9 (Cq), 132.7 (CH), 131.7 (Cq), 129.2(CH), 127.9 (CH), 127.1 (Cq), 126.9 (CH), 126.5 (CH), 125.5 (CH), 123.1 (CH).

HRMS (ESI-QTOF). Calculated for 200.0712 C_12_H_10_NO_2_[M + H]^+^; found, 200.0714.

#### 3.5.3. (*E*)-2-methyl-1-(2-nitrovinyl)naphthalene (**3c**)

Following the general procedure (12 h reaction time, 0.5 eq of ammonium acetate). Obtained as a brown solid. Total of 582.1 mg, yield = 91%.

^1^H NMR (600 MHz, CDCl_3_, 7.26 ppm, TMS, 25 °C) *δ* 8.47 (d, *J* = 13.8 Hz, 1H), 7.93 (d, *J* = 8.3 Hz, 1H), 7.83 (d, *J* = 7.9 Hz, 1H), 7.80 (d, *J* = 8.4 Hz, 1H), 7.53 (ddd, *J* = 8.4, 6.9, 1.5 Hz, 1H), 7.49 (ddd, *J* = 7.9, 6.8, 1.3 Hz, 1H), 7.35 (d, *J* = 8.4 Hz, 1H), 7.32 (d, *J* = 13.8 Hz, 1H), 2.51 (s, 3H).

^13^C{^1^H} NMR (151 MHz, CDCl_3_, 77.16 ppm, TMS, 25 °C) *δ* 141.6 (CH), 136.1 (Cq), 135.9 (CH), 132.1 (Cq), 131.4 (Cq), 130.5 (CH), 129.0 (CH), 128.7 (CH), 127.4 (CH), 125.7 (CH), 125.3 (Cq), 123.8 (CH), 21.3 (CH_3_).

HRMS (ESI-QTOF). Calculated for 214.0668 C_13_H_12_NO_2_ [M + H]^+^; found, 214.0664.

#### 3.5.4. (*E*)-2,3-dimethyl-1-(2-nitrovinyl)naphthalene (**3d**)

Following the general procedure (12 h reaction time, 0.5 eq of ammonium acetate). Obtained as brown needles. Total of 668.1 mg, yield = 98%.

^1^H NMR (600 MHz, CDCl_3_, 7.26 ppm, TMS, 25 °C) *δ* 8.57 (d, *J* = 13.9 Hz, 1H), 7.93–7.86 (m, 1H), 7.81–7.75 (m, 1H), 7.71 (s, 1H), 7.52–7.44 (m, 2H), 7.33 (d, *J* = 13.9 Hz, 1H), 2.48 (s, 3H), 2.45 (s, 3H).

^13^C{^1^H} NMR (151 MHz, CDCl_3_, 77.16 ppm, TMS, 25 °C) *δ* 142.3 (CH), 137.3 (CH), 135.6 (Cq), 135.5 (Cq), 132.2 (Cq), 130.33 (CH), 130.26 (Cq), 128.0 (CH), 126.6 (CH), 125.9 (CH), 125.9 (Cq), 124.1 (CH), 21.2 (CH_3_), 17.7 (CH_3_).

HRMS (ESI-QTOF). Calculated for 228.1025 C_14_H_14_NO_2_ [M + H]^+^; found 228.1028.

#### 3.5.5. (*E*)-2-methoxy-1-(2-nitrovinyl)naphthalene (**3e**)

Following the general procedure (12 h reaction time). Obtained as yellow needles. Total of 673.8 mg, yield = 98%.

^1^H NMR (600 MHz, CDCl_3_, 7.26 ppm, TMS, 25 °C) *δ* 8.17 (d, *J* = 8.7 Hz, 1H), 8.15 (d, *J* = 13.3 Hz, 1H), 7.98 (d, *J* = 9.1 Hz, 1H), 7.83 (d, *J* = 8.1 Hz, 1H), 7.62 (ddd, *J* = 8.5, 6.8, 1.4 Hz, 1H), 7.44 (ddd, *J* = 7.9, 6.8, 0.9 Hz, 1H), 7.32 (d, *J* = 9.1 Hz, 1H), 4.10 (s, 3H).

^13^C{^1^H} NMR (151 MHz, CDCl_3_, 77.16 ppm, TMS, 25 °C) *δ* 159.1 (Cq), 140.3 (CH), 134.5 (CH), 133.5 (Cq), 131.0 (CH), 129.2 (CH), 129.0 (Cq), 128.6 (CH), 124.6 (CH), 122.3 (CH), 112.4 (CH), 111.7 (Cq), 56.4 (CH_3_).

HRMS (ESI-QTOF). Calculated for 230.0817 C_13_H_12_NO_3_ [M + H]^+^; found 230.0815.

#### 3.5.6. (*E*)-2,3-dimethoxy-1-(2-nitrovinyl)naphthalene (**3f**)

Following the general procedure (12 h reaction time). Obtained as a yellow solid. Total of 762.2 mg, yield = 98%.

^1^H NMR (600 MHz, CDCl_3_, 7.26 ppm, TMS, 25 °C) *δ* 8.67 (d, *J* = 13.5 Hz, 1H), 8.06 (d, *J* = 13.5 Hz, 1H), 8.04–8.00 (m, 1H), 7.79–7.72 (m, 1H), 7.52–7.45 (m, 2H), 7.31 (s, 1H), 4.02 (s, 3H), 3.97 (s, 3H).

^13^C{^1^H} NMR (151 MHz, CDCl_3_, 77.16 ppm, TMS, 25 °C) *δ* 151.7 (Cq), 150.8 (Cq), 141.5 (CH), 131.4 (Cq), 131.3 (CH), 127.7 (CH), 127.6 (Cq), 126.2 (CH), 125.8 (CH), 123.0 (CH), 118.7 (Cq), 111.5 (CH), 60.9 (CH_3_), 56.0 (CH_3_).

HRMS (ESI-QTOF). Calculated for 260.0923 C_14_H_14_NO_4_ [M + H]^+^; found 260.0920.

#### 3.5.7. (*E*)-1-(2-nitrovinyl)naphthalen-2-ol (**3g**)

Following the general procedure, with 1.1 eq of ammonium acetate, 100 eq of nitromethane and 12 h reaction time. After the workup, crystallization in DCM was needed for the purification. Product was obtained as a red solid. Total of 477.8 mg, yield = 74%

^1^H NMR (600 MHz, DMSO-d6, 2.50 ppm, TMS, 25 °C) *δ* 11.74 (s, 1H), 8.79 (d, *J* = 13.2 Hz, 1H), 8.31 (d, *J* = 13.2 Hz, 1H), 8.19 (d, *J* = 8.6 Hz, 1H), 8.00 (d, *J* = 8.9 Hz, 1H), 7.89 (dd, *J* = 8.1, 1.4 Hz, 1H), 7.62 (ddd, *J* = 8.4, 6.8, 1.4 Hz, 1H), 7.43 (ddd, *J* = 8.0, 6.7, 1.0 Hz, 1H), 7.31 (d, *J* = 8.9 Hz, 1H).

^13^C{^1^H} NMR (151 MHz, DMSO-d6, 39.51 ppm, TMS, 25 °C) *δ* 158.95 (Cq), 138.80 (CH), 134.65 (CH), 132.98 (Cq), 131.40 (CH), 129.05 (CH), 128.35 (CH), 128.00 (Cq), 123.75 (CH), 121.66 (CH), 118.09 (CH), 108.09 (Cq).

HRMS (ESI-QTOF). Calculated for 216.0661 C_12_H_10_NO_3_ [M + H]^+^; found 216.0658.

#### 3.5.8. (*E*)-2-(benzyloxy)-1-(2-nitrovinyl)naphthalene (**3h**)

Following the general procedure (2 h reaction time). Obtained as a yellow solid. Total of 906.8 mg, yield = 99%

^1^H NMR (600 MHz, CDCl_3_, 7.26 ppm, TMS, 25 °C) *δ* 8.86 (d, *J* = 13.4 Hz, 1H), 8.19 (d, *J* = 8.6 Hz, 1H), 8.13 (d, *J* = 13.4 Hz, 1H), 7.93 (d, *J* = 9.1 Hz, 1H), 7.82 (d, *J* = 7.8 Hz, 1H), 7.63 (ddd, *J* = 8.5, 6.8, 1.4 Hz, 1H), 7.50–7.36 (m, 6H), 7.35 (d, *J* = 9.1 Hz, 1H), 5.40 (s, 2H).

^13^C{^1^H} NMR (151 MHz, CDCl_3_, 77.16 ppm, TMS, 25 °C) *δ* 158.0 (Cq), 140.6 (CH), 135.8 (Cq), 134.3 (CH), 133.5 (Cq), 131.0 (CH), 129.2 (Cq), 129.1 (CH), 129.1 (CH), 128.8 (CH), 128.6 (CH), 127.6 (CH), 124.7 (CH), 122.4 (CH), 113.8 (CH), 112.3 (Cq), 71.6 (CH_2_).

HRMS (ESI-QTOF). Calculated for 306.1130 C_19_H_16_NO_3_ [M + H]^+^; found 306.1134.

### 3.6. General Procedure for Cycloaddition to Afford Pyrrole *(**4**)*

To the enamine solution 2 (0.11 g, 0.5 mmol, 1 eq) in ethanol (5 mL, 0.1 M), nitrovinyl-aryl 3 (0.5 mmol 1 eq) and ZrCp_2_Cl_2_ (14.6 mg, 0.1 eq) were added. The solution was heated to 90 °C in a closed flask for 4h, then it was concentrated in vacuo. The crude solid was treated over a chromatography column with a mixture of Hex/DCM/EtOAc eluent to obtain the product **4** as a pale-yellow solid.

#### 3.6.1. 2-Isopropyl-1-methyl-*N*-phenyl-4-(o-tolyl)-1*H*-pyrrole-3-carboxamide (**4a**)

Eluent Hex/DCM/EtOAc 75:20:5. Obtained 71.5 mg, yield = 43%

^1^H NMR (600 MHz, CD_3_CN, 1.96 ppm, 25 °C) *δ* 7.32–7.22 (m, 5H), 7.22–7.17 (m, 2H), 7.14–7.08 (m, 2H), 6.99 (tt, *J* = 7.4, 1.2 Hz, 1H), 6.52 (s, 1H), 3.76–3.67 (m, 4H), 2.20 (s, 3H), 1.41 (d, *J* = 7.2 Hz, 6H).

^13^C{^1^H} NMR (151 MHz, CD_3_CN, s 118.26 ppm, 25 °C) *δ* 165.6 (Cq), 142.2 (Cq), 140.0 (Cq), 138.3 (Cq), 135.9 (Cq), 131.8 (CH), 131.1 (CH), 129.6 (CH), 128.3 (CH), 126.8 (CH), 124.1 (CH), 122.0 (Cq), 121.7 (CH), 120.0 (CH), 116.4 (Cq), 35.5 (CH_3_), 26.3 (CH), 21.2 (CH_3_), 20.6 (CH_3_).

HRMS (ESI-QTOF). Calculated for 333.1967 C_22_H_25_N_2_O [M + H]^+^; found 333.1969.

#### 3.6.2. 2-Isopropyl-1-methyl-4-(naphthalen-1-yl)-*N*-phenyl-1*H*-pyrrole-3-carboxamide (**4b**)

Eluent Hex/DCM/EtOAc 75:22:3. Obtained 112.4 mg, yield = 61%

^1^H NMR (600 MHz, CD_3_CN, 1.96 ppm, 25 °C) *δ* 8.02–7.96 (m, 1H), 7.93–7.86 (m, 2H), 7.56–7.49 (m, 2H), 7.49–7.41 (m, 2H), 7.33 (s, 1H), 7.10–7.03 (m, 2H), 6.92–6.87 (m, 1H), 6.87–6.83 (m, 2H), 6.68 (s, 1H), 3.79–3.64 (m, 4H), 1.46 (d, *J* = 7.2 Hz, 6H).

^13^C{^1^H} NMR (151 MHz, CD_3_CN, s 118.26 ppm, 25 °C) *δ* 165.6 (Cq), 142.1 (Cq), 139.7 (Cq), 134.7 (Cq), 134.2 (Cq), 133.8 (Cq), 129.4 (CH), 129.1 (CH), 128.9 (CH), 128.4 (CH), 127.1 (CH), 126.9 (CH), 126.9 (CH), 126.5 (CH), 124.1 (CH), 122.6 (CH), 120.6 (Cq), 120.0 (CH), 117.4 (Cq), 35.5 (CH_3_), 26.5 (CH), 21.3 (CH_3_).

HRMS (ESI-QTOF). Calculated for 369.1967 C_25_H_25_N_2_O [M + H]^+^; found 369.1960.

#### 3.6.3. 2-Isopropyl-1-methyl-4-(2-methylnaphthalen-1-yl)-*N*-phenyl-1*H*-pyrrole-3-carboxamide (**4c**)

Eluent Hex/DCM/EtOAc 75:20:5. Obtained 76.5 mg, yield = 40%

^1^H NMR (600 MHz, CD_3_CN, 1.96 ppm, 25 °C) *δ* 7.93–7.86 (m, 2H), 7.70 (ddd, *J* = 7.5, 2.2, 0.9 Hz, 1H), 7.53 (d, *J* = 8.4 Hz, 1H), 7.48–7.40 (m, 2H), 7.08–6.98 (m, 3H), 6.86 (tt, *J* = 7.6, 1.2 Hz, 1H), 6.71–6.65 (m, 2H), 6.53 (s, 1H), 3.96 (eptet, *J* = 7.2 Hz, 1H), 3.77 (s, 3H), 2.36 (s, 3H), 1.48 (dd, *J* = 7.2, 1.9 Hz, 6H).

^13^C{^1^H} NMR (151 MHz, CD_3_CN, s 118.26 ppm, 25 °C) *δ* 164.9 (Cq), 143.9 (Cq), 139.7 (Cq), 137.0 (Cq), 135.1 (Cq), 133.2 (Cq), 132.2 (Cq), 129.7 (CH), 129.5 (CH), 128.9 (CH), 128.8 (CH), 127.4 (CH), 126.7 (CH), 126.2 (CH), 123.8 (CH), 122.6 (CH), 119.4 (CH), 118.4 (Cq), 116.0 (Cq), 35.9 (CH_3_), 26.3 (CH), 21.1 (CH_3_), 21.0 (CH_3_).

HRMS (ESI-QTOF). Calculated for 383.2123 C_26_H_27_N_2_O [M + H]^+^; found 383.2127.

#### 3.6.4. 4-(2,3-Dimethylnaphthalen-1-yl)-2-isopropyl-1-methyl-*N*-phenyl-1*H*-pyrrole-3-carboxamide (**4d**)

Eluent Hex/EtOAc 10:1. Obtained 57.3 mg, yield = 29%

^1^H NMR (600 MHz, CD_3_CN, 1.96 ppm, 25 °C) *δ* 7.85–7.80 (m, 1H), 7.78 (s, 1H), 7.61 (dq, *J* = 9.2, 0.9 Hz, 1H), 7.42 (ddd, *J* = 8.2, 6.7, 1.3 Hz, 1H), 7.37 (ddd, *J* = 8.2, 6.7, 1.4 Hz, 1H), 7.06 (s, 1H), 7.05–7.01 (m, 2H), 6.89–6.84 (m, 1H), 6.69–6.64 (m, 2H), 6.50 (s, 1H), 3.99 (eptet, *J* = 7.2 Hz, 1H), 3.77 (s, 3H), 2.50 (s, 3H), 2.29 (s, 3H), 1.48 (dd, *J* = 7.2, 2.4 Hz, 6H).

^13^C{^1^H} NMR (151 MHz, CD_3_CN, s 118.26 ppm, 25 °C) *δ* 164.8 (Cq), 144.1 (Cq), 139.7 (Cq), 137.2 (Cq), 136.9 (Cq), 133.8 (Cq), 133.2 (Cq), 132.2 (Cq), 129.5 (CH), 128.6 (CH), 128.1 (CH), 126.8 (CH), 126.5 (CH), 126.3 (CH), 123.8 (CH), 122.6 (CH), 119.4 (Cq), 119.1 (CH), 116.1 (Cq), 35.9 (CH_3_), 26.3 (CH), 21.2 (CH_3_), 21.0 (CH_3_), 20.9 (CH_3_), 18.0 (CH_3_).

HRMS (ESI-QTOF). Calculated for 397.2280 C_27_H_29_N_2_O [M + H]^+^; found, 397.2276.

#### 3.6.5. 2-Isopropyl-4-(2-methoxynaphthalen-1-yl)-1-methyl-*N*-phenyl-1*H*-pyrrole-3-carboxamide (**4e**)

Eluent Hex/DCM/EtOAc 65:20:15. Obtained 107.6 mg, yield = 54%

^1^H NMR (600 MHz, CD_3_CN, 1.96 ppm, 25 °C) *δ* 7.96 (d, *J* = 9.1 Hz, 1H), 7.88–7.82 (m, 1H), 7.69 (s, 1H), 7.65 (dq, *J* = 8.6, 0.8 Hz, 1H), 7.50 (d, *J* = 9.1 Hz, 1H), 7.41 (ddd, *J* = 8.5, 6.7, 1.4 Hz, 1H), 7.36 (ddd, *J* = 8.0, 6.7, 1.3 Hz, 1H), 7.14–7.08 (m, 2H), 7.01–6.96 (m, 2H), 6.91 (tt, *J* = 7.3, 1.2 Hz, 1H), 6.52 (s, 1H), 3.90 (s, 3H), 3.78–3.70 (m, 4H), 1.47 (dd, *J* = 7.2, 5.5 Hz, 6H).

^13^C{^1^H} NMR (151 MHz, CD_3_CN, s 118.26 ppm, 25 °C) *δ* 165.4 (Cq), 155.7 (Cq), 142.40 (Cq), 140.0 (Cq), 135.8 (Cq), 130.5 (CH), 130.1 (Cq), 129.5 (CH), 128.8 (CH), 127.6 (CH), 126.1 (CH), 124.8 (CH), 123.8 (CH), 123.1 (CH), 119.5 (CH), 118.9 (Cq), 117.8 (Cq), 115.2 (Cq), 114.5 (CH), 57.2 (CH_3_), 35.6 (CH_3_), 26.5 (CH), 21.5 (CH_3_), 20.9 (CH_3_).

HRMS (ESI-QTOF). Calculated for 399.2073 C_26_H_27_N_2_O_2_ [M + H]^+^; found 399.2078.

#### 3.6.6. 4-(2,3-Dimethoxynaphthalen-1-yl)-2-isopropyl-1-methyl-*N*-phenyl-1*H*-pyrrole-3-carboxamide (**4f**)

Eluent Hex/DCM/EtOAc 65:20:15. Obtained 143.5 mg, yield = 67%

^1^H NMR (600 MHz, CD_3_CN, 1.96 ppm, 25 °C) *δ* 7.91 (s, 1H), 7.82–7.74 (m, 1H), 7.52 (ddt, *J* = 8.5, 1.4, 0.7 Hz, 1H), 7.42–7.37 (m, 2H), 7.29 (ddd, *J* = 8.3, 6.8, 1.3 Hz, 1H), 7.15–7.08 (m, 2H), 7.06–6.99 (m, 2H), 6.91 (tt, *J* = 7.1, 1.2 Hz, 1H), 6.59 (s, 1H), 4.00 (s, 3H), 3.77 (s, 3H), 3.76 (s, 3H), 3.72 (eptet, *J* = 7.2 Hz, 1H), 1.47 (dd, *J* = 8.7, 7.2 Hz, 6H).

^13^C{^1^H} NMR (151 MHz, CD_3_CN, s 118.26 ppm, 25 °C) *δ* 165.3 (Cq), 153.2 (Cq), 148.1 (Cq), 142.3 (Cq), 140.0 (Cq), 132.4 (Cq), 130.8 (Cq), 129.6 (CH), 127.6 (CH), 126.6 (Cq), 126.5 (CH), 126.4 (CH), 125.1 (CH), 123.9 (CH), 122.9 (CH), 119.4 (CH), 117.8 (Cq), 114.7 (Cq), 108.5 (CH), 61.6 (CH_3_), 56.4 (CH_3_), 35.6 (CH_3_), 26.5 (CH), 21.7 (CH_3_), 20.8 (CH_3_).

HRMS (ESI-QTOF). Calculated for 429.2178 C_27_H_29_N_2_O_3_ [M + H]^+^; found 429.2175.

#### 3.6.7. 4-(2-(Benzyloxy)naphthalen-1-yl)-2-isopropyl-1-methyl-*N*-phenyl-1*H*-pyrrole-3-carboxamide (**4h**)

Eluent Hex/DCM/EtOAc 70:25:5. Obtained 138.4 mg, yield = 50%

^1^H NMR (600 MHz, CD_3_CN, 1.96 ppm, 25 °C) *δ* 7.94 (d, *J* = 8.7 Hz, 1H), 7.87 (d, *J* = 8.2 Hz, 1H), 7.71 (ddt, *J* = 8.5, 1.3, 0.8 Hz, 1H), 7.69 (s, 1H), 7.51 (d, *J* = 9.0 Hz, 1H), 7.44 (ddd, *J* = 8.4, 6.7, 1.4 Hz, 1H), 7.39 (ddd, *J* = 8.0, 6.8, 1.3 Hz, 1H), 7.34–7.24 (m, 5H), 7.11–7.05 (m, 2H), 6.92–6.86 (m, 3H), 6.54 (s, 1H), 5.21 (s, 2H), 3.80 (eptet, *J* = 7.3 Hz, 1H), 3.76 (s, 3H), 1.48 (dd, *J* = 7.2, 2.9 Hz, 6H).

^13^C{^1^H} NMR (151 MHz, CD_3_CN, s 118.26 ppm, 25 °C) *δ* 165.3 (Cq), 154.9 (Cq), 142.8 (Cq), 139.9 (Cq), 138.2 (Cq), 135.8 (Cq), 130.5 (Cq), 130.4 (CH), 129.5 (CH), 129.3 (CH), 128.9 (CH), 128.8 (CH), 128.4 (CH), 127.7 (CH), 126.1 (CH), 125.2 (CH), 123.8 (CH), 123.1 (CH), 120.8 (Cq), 119.4 (CH), 117.5 (Cq), 117.2 (CH), 72.6 (CH_2_), 35.7 (CH_3_), 26.5 (CH), 21.5 (CH_3_), 21.0 (CH_3_).

HRMS (ESI-QTOF). Calculated for 475.2386 C_32_H_31_N_2_O_2_ [M + H]^+^; found 475.2392.

### 3.7. General Procedure for Bromination of Pyrrole *(**4**)* to Afford Pyrrole *(**5**)*

In a round-bottom flask equipped with a magnetic stirring bar and under N_2_ atmosphere, the pyrrole **4** (0.20 mmol, 1 eq) was dissolved in anhydrous THF (2 mL, 0.1 M) and cooled to −95 °C. Then, *N*-bromosuccinimide (39.2 mg, 0.22 mmol, 1.1 eq) was added and the solution stirred, avoiding light, for 30 min. Then, the mixture was allowed to warm to room temperature for 1 h, still stirring and avoiding light. Hence, the solution was diluted with EtOAc, consequently passed in a silica plug and washed with a saturated solution of NaHCO_3_ (3 × 10 mL). The organic layer was dried over Na_2_SO_4_ and concentrated in vacuo, resulting in the product **5** as a pale-yellow solid. No further purification was needed.

#### 3.7.1. 5-Bromo-2-isopropyl-1-methyl-*N*-phenyl-4-(o-tolyl)-1*H*-pyrrole-3-carboxamide (**5a**)

Obtained 82.3 mg as a pale-yellow solid.

^1^H NMR (600 MHz, CD_3_CN, 1.96 ppm, 25 °C) *δ* 7.37–7.25 (m, 5H), 7.23–7.17 (m, 2H), 7.12–7.06 (m, 2H), 7.00 (tt, *J* = 7.3, 1.2 Hz, 1H), 3.83–3.75 (m, 1H), 3.71 (s, 3H), 2.15 (s, 3H), 1.41 (dd, *J* = 7.2, 2.6 Hz, 6H).

^13^C{^1^H} NMR (151 MHz, CD_3_CN, s 118.26 ppm, 25 °C) *δ* 164.6 (Cq), 143.1 (Cq), 139.7 (Cq), 139.1 (Cq), 135.0 (Cq), 132.2 (CH), 131.2 (CH), 129.6 (CH), 129.3 (CH), 127.0 (CH), 124.4 (CH), 122.1 (Cq), 120.0 (CH), 117.2 (Cq), 103.2 (Cq), 34.3 (CH_3_), 27.1 (CH), 21.3 (CH_3_), 20.9 (CH_3_), 20.1 (CH_3_).

HRMS (ESI-QTOF). Calculated for 411.1072 C_22_H_24_BrN_2_O [M + H]^+^; found 411.1076.

#### 3.7.2. 5-Bromo-2-isopropyl-1-methyl-4-(naphthalen-1-yl)-*N*-phenyl-1*H*-pyrrole-3-carboxamide (**5b**)

Obtained 89.5 mg as a pale-yellow fluffy solid.^1^H NMR (600 MHz, CD_3_CN, 1.96 ppm, 25 °C) *δ* 7.95 (ddt, *J* = 7.6, 5.8, 1.1 Hz, 2H), 7.77–7.69 (m, 1H), 7.60 (dd, *J* = 8.3, 7.0 Hz, 1H), 7.56–7.45 (m, 3H), 7.41 (s, 1H), 7.11–7.03 (m, 2H), 6.94–6.88 (m, 1H), 6.88–6.82 (m, 2H), 3.87–3.64 (m, 4H), 1.46 (dd, *J* = 7.2, 4.7 Hz, 6H).

^13^C{^1^H} NMR (151 MHz, CD_3_CN, s 118.26 ppm, 25 °C) *δ* 164.7 (Cq), 142.8 (Cq), 139.4 (Cq), 134.7 (Cq), 133.5 (Cq), 133.2 (Cq), 130.0 (CH), 129.5 (CH), 129.27 (CH), 129.26 (Cq), 127.4 (CH), 127.1 (CH), 126.64 (CH), 126.56 (CH), 124.3 (CH), 120.7 (Cq), 119.9 (CH), 118.6 (Cq), 104.1 (Cq), 34.3 (CH_3_), 27.3 (CH), 21.4 (CH_3_), 21.0 (CH_3_).

HRMS (ESI-QTOF). Calculated for 447.1072 C_25_H_24_BrN_2_O [M + H]^+^; found 447.1068.

#### 3.7.3. 5-Bromo-2-isopropyl-1-methyl-4-(2-methylnaphthalen-1-yl)-*N*-phenyl-1*H*-pyrrole-3-carboxamide (**5c**)

Obtained 92.3 mg as a pale-yellow solid.

^1^H NMR (600 MHz, CD_3_CN, 1.96 ppm, 25 °C) *δ* 7.97–7.90 (m, 2H), 7.58 (ddt, *J* = 6.5, 2.7, 0.8 Hz, 1H), 7.55 (d, *J* = 8.5 Hz, 1H), 7.51–7.45 (m, 2H), 7.14 (s, 1H), 7.08–7.02 (m, 2H), 6.91–6.85 (m, 1H), 6.74–6.69 (m, 2H), 3.97 (eptet, *J* = 7.2 Hz, 1H), 3.78 (s, 3H), 2.33 (s, 3H), 1.47 (dd, *J* = 7.2, 2.7 Hz, 6H).

^13^C{^1^H} NMR (151 MHz, CD_3_CN, s 118.26 ppm, 25 °C) *δ* 164.1 (Cq), 144.5 (Cq), 139.4 (Cq), 137.5 (Cq), 134.2 (Cq), 133.2 (Cq), 131.0 (Cq), 129.7 (CH), 129.5 (CH), 129.4 (CH), 129.1 (CH), 127.8 (CH), 126.4 (CH), 126.1 (CH), 124.2 (CH), 119.5 (CH), 119.1 (Cq), 117.0 (Cq), 104.4 (Cq), 34.6 (CH_3_), 27.1 (CH), 21.0 (CH_3_), 20.9_7_ (CH_3_), 20.7 (CH_3_).

HRMS (ESI-QTOF). Calculated for 461.1229 C_26_H_26_BrN_2_O [M + H]^+^; found 461.1223.

#### 3.7.4. 5-Bromo-4-(2,3-dimethylnaphthalen-1-yl)-2-isopropyl-1-methyl-*N*-phenyl-1*H*-pyrrole-3-carboxamide (**5d**)

Obtained 95.1 mg as a pale-yellow solid.

^1^H NMR (600 MHz, CD_3_CN, 1.96 ppm, 25 °C) *δ* 7.95 (ddt, *J* = 7.6, 5.8, 1.1 Hz, 2H), 7.76–7.72 (m, 1H), 7.60 (dd, *J* = 8.3, 7.0 Hz, 1H), 7.55–7.46 (m, 3H), 7.41 (s, 1H), 7.10–7.04 (m, 2H), 6.93–6.88 (m, 1H), 6.87–6.81 (m, 2H), 3.79–3.68 (m, 4H), 1.46 (dd, *J* = 7.2, 4.7 Hz, 6H).

^13^C{^1^H} NMR (151 MHz, CD_3_CN, s 118.26 ppm, 25 °C) *δ* 164.1 (Cq), 144.7 (Cq), 139.4 (Cq), 137.6 (Cq), 137.0 (Cq), 133.3 (Cq), 132.9 (Cq), 131.1 (Cq), 129.5 (CH), 129.2 (CH), 128.4 (CH), 126.9 (CH), 126.5 (CH), 126.2 (CH), 124.2 (CH), 119.7 (Cq), 119.5 (CH), 117.0 (Cq), 104.6 (Cq), 34.6 (CH_3_), 27.1 (CH), 21.1 (CH_3_), 21.0 (CH_3_), 20.9 (CH_3_), 17.6 (CH_3_).

HRMS (ESI-QTOF). Calculated for 475.1385 C_27_H_28_BrN_2_O [M + H]^+^; found, 475.1381.

#### 3.7.5. 5-Bromo-2-isopropyl-4-(2-methoxynaphthalen-1-yl)-1-methyl-*N*-phenyl-1*H*-pyrrole-3-carboxamide (**5e**)

Obtained 95.5 mg as a pale-yellow solid.

^1^H NMR (600 MHz, CD_3_CN, 1.96 ppm, 25 °C) *δ* 8.02 (d, *J* = 9.1 Hz, 1H), 7.89 (d, *J* = 8.1 Hz, 1H), 7.78 (s, 1H), 7.54 (d, *J* = 9.1 Hz, 1H), 7.47–7.41 (m, 2H), 7.38 (ddd, *J* = 8.0, 5.9, 2.1 Hz, 1H), 7.16–7.09 (m, 2H), 7.03–6.98 (m, 2H), 6.93 (tt, *J* = 7.3, 1.1 Hz, 1H), 3.96 (s, 3H), 3.83–3.69 (m, 4H), 1.47 (dd, *J* = 7.2, 5.6 Hz, 6H).

^13^C{^1^H} NMR (151 MHz, CD_3_CN, s 118.26 ppm, 25 °C) *δ* 164.6 (Cq), 156.0 (Cq), 143.3 (Cq), 139.6 (Cq), 134.9 (Cq), 131.3 (CH), 130.1 (Cq), 129.6 (CH), 129.1 (CH), 128.0 (CH), 125.6 (CH), 125.0 (CH), 124.2 (CH), 119.6 (CH), 118.6 (Cq), 117.8 (Cq), 116.3 (Cq), 114.7 (CH), 104.7 (Cq), 57.4 (CH_3_), 34.4 (CH_3_), 27.3 (CH), 21.6 (CH_3_), 20.8 (CH_3_).

HRMS (ESI-QTOF). Calculated for 477.1178 C_26_H_26_BrN_2_O_2_ [M + H]^+^; found 477.1184.

#### 3.7.6. 5-Bromo-4-(2,3-dimethoxynaphthalen-1-yl)-2-isopropyl-1-methyl-*N*-phenyl-1*H*-pyrrole-3-carboxamide (**5f**)

Obtained 101.5 mg as a pale-yellow solid.

^1^H NMR (600 MHz, CD_3_CN, 1.96 ppm, 25 °C) *δ* 8.16 (s, 1H), 7.85–7.78 (m, 1H), 7.46–7.38 (m, 2H), 7.36–7.29 (m, 2H), 7.15 (tt, *J* = 7.1, 1.8 Hz, 2H), 7.13–7.08 (m, 2H), 6.94 (tt, *J* = 7.0, 1.4 Hz, 1H), 4.01 (s, 3H), 3.80 (s, 3H), 3.76 (s, 3H), 3.70 (eptet, *J* = 7.2 Hz, 1H), 1.46 (dd, *J* = 7.2, 6.1 Hz, 6H).

^13^C{^1^H} NMR (151 MHz, CD_3_CN, s 118.26 ppm, 25 °C) *δ* 164.63 (Cq), 153.2 (Cq), 148.4 (Cq), 143.1 (Cq), 139.8 (Cq), 132.6 (Cq), 129.7 (CH), 129.6 (Cq), 127.9 (CH), 126.7 (CH), 126.0 (CH), 125.6 (Cq), 125.5 (CH), 124.2 (CH), 119.5 (CH), 118.7 (Cq), 115.9 (Cq), 109.2 (CH), 104.5 (Cq), 61.6 (CH_3_), 56.5 (CH_3_), 34.3 (CH_3_), 27.4 (CH), 21.8 (CH_3_), 20.7 (CH_3_).

HRMS (ESI-QTOF). Calculated for 507.1283 C_27_H_28_BrN_2_O_3_ [M + H]^+^; found 507.1278.

#### 3.7.7. 4-(2-(Benzyloxy)naphthalen-1-yl)-5-bromo-2-isopropyl-1-methyl-*N*-phenyl-1*H*-pyrrole-3-carboxamide (**5h**)

Obtained 110.7 mg as a pale-yellow solid.

^1^H NMR (600 MHz, CDCl_3_, 7.26 ppm, TMS, 25 °C) *δ* 7.90 (d, *J* = 8.9 Hz, 1H), 7.83 (dd, *J* = 8.0, 1.5 Hz, 1H), 7.81 (s, 1H), 7.57 (dq, *J* = 8.4, 0.9 Hz, 1H), 7.43 (ddd, *J* = 8.4, 6.7, 1.4 Hz, 1H), 7.41–7.36 (m, 2H), 7.28–7.20 (m, 5H), 7.09–7.02 (m, 2H), 6.87 (ddt, *J* = 14.5, 7.3, 1.2 Hz, 3H), 5.19 (d, *J* = 12.2 Hz, 1H), 5.14 (d, *J* = 12.2 Hz, 1H), 4.15 (eptet, *J* = 7.3 Hz, 1H), 3.79 (s, 3H), 1.52 (dd, *J* = 7.3, 5.4 Hz, 6H).

^13^C{^1^H} NMR (151 MHz, CDCl_3_, 77.16 ppm, TMS, 25 °C) *δ* 163.6 (Cq), 154.5 (Cq), 143.9 (Cq), 138.7 (Cq), 137.1 (Cq), 134.1 (Cq), 130.6 (CH), 129.9, 128.7 (CH), 128.6 (CH), 128.2 (CH), 127.9 (CH), 127.4 (CH), 127.2 (CH), 125.4 (CH), 124.7 (CH), 123.2 (CH), 119.4 (Cq), 119.1 (CH), 117.1 (Cq), 116.7 (CH), 115.5 (Cq), 104.7 (Cq), 72.5 (CH_2_), 34.3 (CH_3_), 26.2 (CH), 21.3 (CH_3_), 20.9 (CH_3_).

HRMS (ESI-QTOF). Calculated for 553.1491 C_32_H_30_BrN_2_O_2_ [M + H]^+^; found 553.1497.

### 3.8. General Procedure to Obtain Pyrrole *(**1**)*

In a two-neck round bottom flask equipped with a magnetic stirring bar, reflux condenser and under N_2_ atmosphere, a solution of compound **5** (0.052 mmol, 1 eq) in toluene/CPME (19:1, 0.17 mL), phenylboronic acid (9.51 mg, 0.078 mmol, 1.5 eq) and a solution of K_2_CO_3_ (2 M, 62.4 μL, 2.5 eq) were added. The resulting mixture was degassed with cycles of vacuum/nitrogen. Then, a catalytic amount of Pd(PPh_3_)_4_ was added over nitrogen flow and the resulting mixture refluxed for 4 h, avoiding light. The solution was passed through a Celite plug, then concentrated in vacuo. The resulting crude was dissolved in DCM, washed with water and extracted with DCM. The combined organic layer was dried over Na_2_SO_4_, concentrated in vacuo, and the product was purified by column chromatography with a mixture of hexane\DCM\EtOAc, obtaining **1** as a white solid.

#### 3.8.1. 2-Isopropyl-1-methyl-*N*,5-diphenyl-4-(o-tolyl)-1*H*-pyrrole-3-carboxamide (**1a**)

Hexane/DCM/EtOAc 75:20:5. Obtained 15.5 mg, yield = 73%

^1^H NMR (600 MHz, C_4_D_2_Cl_4_, 6 ppm, 25 °C) *δ* 7.11–7.30 (m, 11H), 7.01 (s, 1H), 6.94–6.99 (m, 3H), 4.25 (eptet, *J* = 7.2 Hz, 1H), 3.57 (s, 3H), 2.07 (s, 3H), 1.51 (d, *J* = 7.2 Hz, 3H), 1.50 (d, *J* = 7.22 Hz, 3H).

^13^C{^1^H} NMR (151 MHz, C_4_D_2_Cl_4_, 73.78 ppm, 25 °C) *δ* 164.0 (Cq), 143.5 (Cq), 138.6 (Cq), 138.5 (Cq), 134.8 (Cq), 131.9 (Cq), 131.7 (Cq), 130.9 (Cq), 130.5 (CH), 130.2 (CH), 128.5 (CH), 127.9 (CH), 127.7 (CH), 127.1 (CH), 126.0 (CH), 122.9 (CH), 119.4 (Cq), 119.2 (CH), 113.4 (Cq), 33.4 (CH_3_), 25.2 (CH), 20.8 (CH_3_), 20.7 (CH_3_), 20.1 (CH_3_).

HRMS (ESI-QTOF). Calculated for 409.2280 C_28_H_29_N_2_O [M + H]^+^; found 409.2274.

#### 3.8.2. 2-Isopropyl-1-methyl-4-(naphthalen-1-yl)-*N*,5-diphenyl-1*H*-pyrrole-3-carboxamide (**1b**)

Hexane/DCM/EtOAc 75:20:5. Obtained 19.2 mg, yield = 83%

^1^H NMR (600 MHz, CD_3_CN, 1.96 ppm, 25 °C) *δ* 7.88–7.84 (m, 1H), 7.84–7.78 (m, 2H), 7.49–7.38 (m, 4H), 7.26 (s, 1H), 7.22–7.13 (m, 5H), 7.07–7.03 (m, 2H), 6.90–6.84 (m, 1H), 6.81–6.75 (m, 2H), 3.80 (eptet, *J* = 7.2 Hz, 1H), 3.59 (s, 3H), 1.52 (dd, *J* = 9.7, 7.2 Hz, 6H).

^13^C{^1^H} NMR (151 MHz, CD_3_CN, s 118.26 ppm, 25 °C) *δ* 165.4 (Cq), 142.3 (Cq), 139.6 (Cq), 134.5 (Cq), 134.5 (Cq), 134.5 (Cq), 133.0 (Cq), 132.6 (Cq), 131.6 (CH), 130.4 (CH), 129.4 (CH), 129.1 (CH), 128.9 (CH), 128.6 (CH), 128.5 (CH), 127.3 (CH), 127.0 (CH), 126.9 (CH), 126.3 (CH), 124.1 (CH), 119.7 (CH), 119.1 (Cq), 117.5 (Cq), 33.4 (CH_3_), 26.8 (CH), 21.5 (CH_3_), 21.1 (CH_3_).

HRMS (ESI-QTOF). Calculated for 445.2280 C_31_H_29_N_2_O [M + H]^+^; found 445.2276.

#### 3.8.3. 2-Isopropyl-1-methyl-4-(2-methylnaphthalen-1-yl)-*N*,5-diphenyl-1*H*-pyrrole-3-carboxamide (**1c**)

Hexane/DCM/EtOAc 75:20:5. Obtained 19.1 mg, yield = 80%

^1^H NMR (600 MHz, CD_3_CN, 1.96 ppm, 25 °C) *δ* 7.86 (dd, *J* = 7.6, 1.7 Hz, 1H), 7.80 (d, *J* = 8.4 Hz, 1H), 7.78–7.74 (m, 1H), 7.45 (dddd, *J* = 16.4, 8.1, 6.8, 1.5 Hz, 2H), 7.39 (d, *J* = 8.5 Hz, 1H), 7.19–7.13 (m, 3H), 7.13–7.08 (m, 2H), 7.06 (s, 1H), 7.05–7.01 (m, 2H), 6.88–6.84 (m, 1H), 6.70–6.65 (m, 2H), 4.00 (eptet, *J* = 7.2 Hz, 1H), 3.61 (s, 3H), 2.24 (s, 3H), 1.54 (dd, *J* = 7.2, 1.0 Hz, 6H).

^13^C{^1^H} NMR (151 MHz, CD_3_CN, s 118.26 ppm, 25 °C) *δ* 165.0 (Cq), 144.0 (Cq), 139.6 (Cq), 137.3 (Cq), 135.1 (Cq), 133.2 (Cq), 133.0 (Cq), 132.4 (Cq), 132.3 (Cq), 131.0 (CH), 129.5 (CH), 129.5 (CH), 129.0 (CH), 129.0 (CH), 128.8 (CH), 128.4 (CH), 127.6 (CH), 126.7 (CH), 126.2 (CH), 123.9 (CH), 119.4 (CH), 117.3 (Cq), 115.9 (Cq), 33.7 (CH_3_), 26.7 (CH), 21.2 (CH_3_), 21.0_3_ (CH_3_), 20.9_9_ (CH_3_).

HRMS (ESI-QTOF). Calculated for 459.2436 C_32_H_31_N_2_O [M + H]^+^; found 459.2429.

#### 3.8.4. 4-(2,3-Dimethylnaphthalen-1-yl)-2-isopropyl-1-methyl-*N*,5-diphenyl-1*H*-pyrrole-3-carboxamide (**1d**)

Petroleum ether/DCM 1:1. Obtained 18.7 mg, yield = 76%

^1^H NMR (600 MHz, CD_3_CN, 1.96 ppm, 25 °C) *δ* 7.79–7.74 (m, 1H), 7.68 (s, 1H), 7.66 (dd, *J* = 7.7, 1.8 Hz, 1H), 7.43–7.35 (m, 2H), 7.16 (dt, *J* = 4.4, 2.9 Hz, 3H), 7.14–7.10 (m, 2H), 7.07 (s, 1H), 7.06–7.00 (m, 2H), 6.86 (tt, *J* = 7.6, 1.2 Hz, 1H), 6.69–6.63 (m, 2H), 4.03 (eptet, *J* = 7.2 Hz, 1H), 3.60 (s, 3H), 2.40 (s, 3H), 2.22 (s, 3H), 1.54 (dd, *J* = 7.2, 3.0 Hz, 6H).

^13^C{^1^H} NMR (151 MHz, CD_3_CN, s 118.26 ppm, 25 °C) *δ* 165.0 (Cq), 144.1 (Cq), 139.7 (Cq), 137.5 (Cq), 136.8 (Cq), 133.7 (Cq), 133.3 (Cq), 133.1 (Cq), 132.4 (Cq), 132.3 (Cq), 131.0 (CH), 129.5 (CH), 128.9 (CH), 128.6 (CH), 128.5 (CH), 128.3 (CH), 126.8 (CH), 126.7 (CH), 126.3 (CH), 123.9 (CH), 119.3 (CH), 117.8 (Cq), 115.8 (Cq), 33.7 (CH_3_), 26.7 (CH), 21.1 (CH_3_), 21.1 (CH_3_), 21.0 (CH_3_), 18.0 (CH_3_).

HRMS (ESI-QTOF). Calculated for 473.2593 C_33_H_33_N_2_O [M + H]^+^; found 473.2587.

#### 3.8.5. 2-Isopropyl-4-(2-methoxynaphthalen-1-yl)-1-methyl-*N*,5-diphenyl-1*H*-pyrrole-3-carboxamide (**1e**)

Hexane/DCM/EtOAc 40:50:10. Obtained 23.9 mg, yield = 97%.

^1^H NMR (600 MHz, CD_3_CN, 1.96 ppm, 25 °C) *δ* 7.90–7.84 (m, 1H), 7.78 (s, 1H), 7.76–7.72 (m, 1H), 7.46 (d, *J* = 9.1 Hz, 1H), 7.42 (dq, *J* = 8.6, 1.0 Hz, 1H), 7.31 (ddd, *J* = 8.4, 6.7, 1.3 Hz, 1H), 7.25 (ddd, *J* = 8.1, 6.8, 1.3 Hz, 1H), 7.21–7.09 (m, 7H), 7.03–6.98 (m, 2H), 6.92 (tt, *J* = 7.3, 1.2 Hz, 1H), 3.98 (s, 3H), 3.82 (eptet, *J* = 6.9 Hz, 1H), 3.60 (s, 3H), 1.53 (dd, *J* = 8.1, 7.2 Hz, 6H).

^13^C{^1^H} NMR (151 MHz, CD_3_CN, s 118.26 ppm, 25 °C) *δ* 165.5 (Cq), 156.5 (Cq), 142.7 (Cq), 139.9 (Cq), 135.5 (Cq), 133.2 (Cq), 132.7 (Cq), 131.0 (CH), 130.6 (CH), 129.8 (Cq), 129.6 (CH), 128.9 (CH), 128.8 (CH), 128.3 (CH), 127.6 (CH), 125.8 (CH), 124.7 (CH), 123.9 (CH), 119.4 (CH), 119.1 (Cq), 117.7 (Cq), 114.3 (Cq), 114.1 (CH), 57.0 (CH_3_), 33.5 (CH_3_), 26.8 (CH), 22.0 (CH_3_), 20.8 (CH_3_).

HRMS (ESI-QTOF). Calculated for 475.2386 C_32_H_31_N_2_O_2_ [M + H]^+^; found 475.2378.

#### 3.8.6. 4-(2,3-Dimethoxynaphthalen-1-yl)-2-isopropyl-1-methyl-*N*,5-diphenyl-1*H*-pyrrole-3-carboxamide (**1f**)

Hexane/DCM/EtOAc 70:20:10. Obtained 22.6 mg, yield = 86%.

^1^H NMR (600 MHz, CD_3_CN, 1.96 ppm, 25 °C) *δ* 7.75 (s, 1H), 7.71 (ddd, *J* = 8.2, 1.3, 0.6 Hz, 1H), 7.48 (ddd, *J* = 8.3, 1.3, 0.7 Hz, 1H), 7.34 (ddd, *J* = 8.2, 6.9, 1.3 Hz, 1H), 7.29 (s, 1H), 7.27 (ddd, *J* = 8.3, 6.8, 1.3 Hz, 1H), 7.20–7.13 (m, 4H), 7.13–7.07 (m, 2H), 6.98–6.93 (m, 2H), 6.90 (td, *J* = 7.3, 1.2 Hz, 1H), 3.92 (s, 3H), 3.76 (eptet, *J* = 7.2 Hz, 1H), 3.70 (s, 3H), 3.59 (s, 3H), 1.51 (dd, *J* = 7.2, 2.4 Hz, 6H).

^13^C{^1^H} NMR (151 MHz, CD_3_CN, s 118.26 ppm, 25 °C) *δ* 165.4 (Cq), 153.2 (Cq), 148.7 (Cq), 142.6 (Cq), 139.9 (Cq), 133.1 (Cq), 132.4 (Cq), 132.1 (Cq), 131.1 (CH), 130.5 (Cq), 129.6 (CH), 128.9 (CH), 128.4 (CH), 127.7 (CH), 126.4 (CH), 126.4 (CH), 126.2 (Cq), 125.3 (CH), 124.0 (CH), 119.4 (CH), 117.7 (Cq), 114.0 (Cq), 108.6 (CH), 61.2 (CH_3_), 56.3 (CH_3_), 33.5 (CH_3_), 26.9 (CH), 21.6 (CH_3_), 21.0 (CH_3_).

HRMS (ESI-QTOF). Calculated for 505.2491 C_33_H_33_N_2_O_3_ [M + H]^+^; found 505.2490.

#### 3.8.7. 4-(2-(Benzyloxy)naphthalen-1-yl)-2-isopropyl-1-methyl-*N*,5-diphenyl-1*H*-pyrrole-3-carboxamide (**1h**)

Hexane/DCM/EtOAc 70:10:5. CSP-HPLC separation for the evaluation of the rotational energy barrier: AD-H, 60:40 Hex/IPA, 20 mL/min, first elution = 3.9 min, second elution = 11.0 min). Obtained 26.1 mg, yield = 91%.

^1^H NMR (600 MHz, CDCl_3_, 7.26 ppm, TMS, 25 °C) *δ* 7.77–7.70 (m, 3H), 7.44 (s, 1H), 7.38 (ddd, *J* = 8.4, 6.8, 1.3 Hz, 1H), 7.30 (ddd, *J* = 8.0, 6.8, 1.2 Hz, 1H), 7.25–7.19 (m, 4H), 7.15 (dd, *J* = 7.3, 2.4 Hz, 2H), 7.11–7.05 (m, 3H), 7.05–6.99 (m, 4H), 6.83 (tt, *J* = 7.4, 1.2 Hz, 1H), 6.74 (dd, *J* = 8.6, 1.2 Hz, 2H), 5.11 (d, *J* = 12.6 Hz, 1H), 5.02 (d, *J* = 12.6 Hz, 1H), 4.22 (eptet, *J* = 7.3 Hz, 1H), 3.64 (s, 3H), 1.59 (d, *J* = 7.2 Hz, 6H).

^13^C{^1^H} NMR (151 MHz, CDCl_3_, 77.16 ppm, TMS, 25 °C) *δ* 164.5 (Cq), 154.7 (Cq), 143.7 (Cq), 138.8 (Cq), 137.5 (Cq), 135.1 (Cq), 132.3 (Cq), 132.2 (Cq), 130.2 (CH), 129.8 (CH), 129.3 (Cq), 128.5 (CH), 128.5 (CH), 128.0 (CH), 127.9 (CH), 127.6 (CH), 127.21 (CH), 127.19 (CH), 126.7 (CH), 125.6 (CH), 124.2 (CH), 122.9 (CH), 119.6 (Cq), 119.1 (CH), 115.64 (Cq), 115.58 (CH), 113.5 (CH), 71.2 (CH_2_), 33.6 (CH_3_), 25.9 (CH), 21.2 (CH_3_), 21.0 (CH_3_).

HRMS (ESI-QTOF). Calculated for 551.2699 C_38_H_35_N_2_O_2_ [M + H]^+^; found 551.2703.

#### 3.8.8. 4-(2-Hydroxynaphthalen-1-yl)-2-isopropyl-1-methyl-*N*,5-diphenyl-1*H*-pyrrole-3-carboxamide (**1g**)

In a two-neck round-bottom flask equipped with a magnetic stirring bar and under N_2_, the compound **1e** (59 mg, 0.12 mmol, 1 eq) was dissolved in dichloromethane (4.1 mL, 0.03 M) and cooled to 0 °C. Then, a 1 M solution of BBr_3_ in dichloromethane (310 µL, 0.31 mmol, 2.5 eq) was added dropwise. The resulting mixture was then allowed to warm to room temperature and was stirred for 1 h and 30 min. Successively, 4 mL of cool water was added at 0 °C and then stirred for another 20 min. The solution was then extracted with DCM (2 × 10 mL) and the combined organic layer was dried over Na_2_SO_4_ and concentrated in vacuo. The product was purified by column chromatography with a mixture 60:20:20 of hexane\DCM\EtOAc eluent, obtaining **1g** (40.3 mg, 73% yield) as a white solid.

^1^H NMR (600 MHz, CDCl_3_, 7.26 ppm, TMS, 25 °C) *δ* 7.79–7.74 (m, 2H), 7.59 (dd, *J* = 8.4, 1.1 Hz, 1H), 7.40 (ddd, *J* = 8.2, 6.8, 1.3 Hz, 1H), 7.30 (ddd, *J* = 8.1, 6.9, 1.2 Hz, 1H), 7.20–7.12 (m, 5H), 7.11–7.07 (m, 2H), 7.04–6.98 (m, 2H), 6.87–6.82 (m, 1H), 6.67–6.61 (m, 2H), 5.62 (s, 1H, 1H), 4.28 (eptet, *J* = 7.2 Hz, 1H), 3.67 (s, 3H), 1.56 (d, *J* = 7.1 Hz, 6H).

^13^C{^1^H} NMR (151 MHz, CDCl_3_, 77.16 ppm, TMS, 25 °C) *δ* 163.6 (Cq), 152.5 (Cq), 145.6 (Cq), 138.3 (Cq), 134.5 (Cq), 133.8 (Cq), 130.9 (Cq), 130.5 (CH), 130.0 (CH), 129.0 (Cq), 128.6 (CH), 128.5 (CH), 128.4 (CH), 128.1 (CH), 127.6 (CH), 124.5 (CH), 123.8 (CH), 123.3 (CH), 119.3 (CH), 117.4 (CH), 115.2 (Cq), 113.5 (Cq), 110.1 (Cq), 33.9 (CH_3_), 25.9 (CH), 21.1 (CH_3_), 20.8 (CH_3_).

HRMS (ESI-QTOF). Calculated for 461.2229 C_31_H_29_N_2_O_2_ [M + H]^+^; found 461.2229.

### 3.9. T-butyl 2-((4R,6R)-2,2-dimethyl-6-(2-(((Z)-4-methyl-1-oxo-1-(phenylamino)pent-2-en-3-yl)amino)ethyl)-1,3-dioxan-4-yl)acetate *(**2’**)*

In a round-bottom flask equipped with a magnetic stirring bar and a reflux condenser, 4-methyl-3-oxo-*N*-phenylpentanamide (0.72 g, 3.49 mmol), *tert*-butyl 2-((4*R*,6*R*)-6-(2-aminoethyl)-2,2-dimethyl-1,3-dioxan-4-yl)acetate (0.95 g, 3.49 mmol) and Yb(OTf)_3_ (22 mg, 0.04 mmol, 0.01 eq) were dissolved in EtOH (4 mL, 0.9 M). The reaction was conducted at 40 °C for 12 h, and then the mixture was passed through basic alumina eluted with EtOH. Concentrated in vacuo, the resultant pale-yellow oil could be used without further purification. If crystallized in hot ACN, product 2′ could be obtained as a white powder (1.1 g, yield = 65%).

1H NMR (600 MHz, CD_3_CN, 1.96 ppm, 25 °C) *δ* 9.30 (s, 1H), 7.66 (s, 1H), 7.54–7.50 (m, 2H), 7.28–7.23 (m, 2H), 6.97 (tt, *J* = 7.3, 1.2 Hz, 1H), 4.59 (s, 1H), 4.28 (dddd, *J* = 11.5, 7.6, 4.9, 2.5 Hz, 1H), 4.03 (tt, *J* = 11.4, 2.9 Hz, 1H), 3.39–3.28 (m, 2H), 2.78–2.70 (m, 1H), 2.33 (dd, *J* = 14.9, 4.8 Hz, 1H), 2.25 (dd, *J* = 14.9, 8.1 Hz, 1H), 1.72–1.61 (m, 2H), 1.57 (dt, *J* = 12.8, 2.5 Hz, 1H), 1.45 (s, 3H), 1.44 (s, 9H), 1.32 (s, 3H), 1.22–1.09 (m, 7H).

13C{1H} NMR (151 MHz, CD_3_CN, s 118.26 ppm, 25 °C) *δ* 170.9 (Cq), 170.7 (Cq), 141.4 (Cq), 129.5 (CH), 122.8 (CH), 119.5 (CH), 99.5 (Cq), 81.5 (CH), 80.9 (Cq), 67.2 (CH), 67.2 (CH), 43.5 (CH_2_), 38.7 (CH_2_), 38.1 (CH_2_), 37.2 (CH_2_), 30.4 (CH_3_), 29.1 (CH), 28.2 (CH_3_), 21.9 (CH_3_), 21.9 (CH_3_), 20.1 (CH_3_).

HRMS (ESI-QTOF). Calculated for 461.3015 C_26_H_41_N_2_O_5_ [M + H]^+^; found 461.3021.

### 3.10. T-butyl 2-((4R,6R)-6-(2-(2-isopropyl-4-(2-methoxynaphthalen-1-yl)-3-(phenylcarbamoyl)-1H-pyrrol-1-yl)ethyl)-2,2-dimethyl-1,3-dioxan-4-yl)acetate *(**4e’**)*

In a test tube equipped with a magnetic stirring bar, the enamine solution **2′** (0.23 g, 0.5 mmol, 1 eq) in ethanol (5 mL 0.1 M), (*E*)-2-methoxy-1-(2-nitrovinyl)naphthalene **3e** (0.14 g, 0.6 mmol 1.2 eq) and ZrCp_2_Cl_2_ (14.6 mg, 0.1 eq) were added. The vessel was sealed and the solution was heated to 40 °C for 4 h, then it was concentrated in vacuo. The crude solid was treated over the chromatography column with a gradient mixture of Toluene/EtOAc 10:1 to 8:1 to obtain 0.2 g of product **4e’** (yield = 63%) as a pale-yellow solid.

Analyzed as a couple of conformers.

^1^H NMR (600 MHz, CD_3_CN, 1.96 ppm, 25 °C) *δ* 7.95 (d, *J* = 9.1 Hz, 2H), 7.87–7.83 (m, 2H), 7.79 (s, 1H), 7.76 (s, 1H), 7.68 (d, *J* = 8.5 Hz, 1H), 7.65 (d, *J* = 8.5 Hz, 1H), 7.47 (d, *J* = 9.1 Hz, 2H), 7.45–7.38 (m, 2H), 7.38–7.32 (m, 2H), 7.15–7.09 (m, 4H), 7.05–6.97 (m, 4H), 6.94–6.87 (m, 2H), 6.56 (s, 1H), 6.54 (s, 1H), 4.32–4.24 (m, 2H), 4.19–4.13 (m, 1H), 4.12–4.03 (m, 3H), 4.02–3.91 (m, 3H), 3.89 (s, 6H), 3.61–3.48 (m, 3H), 2.35 (dd, *J* = 6.4, 4.8 Hz, 1H), 2.33 (dd, *J* = 6.4, 4.7 Hz, 1H), 2.28 (d, *J* = 8.3 Hz, 1H), 2.25 (d, *J* = 8.2 Hz, 1H), 2.00–1.90 (m, 2H), 1.89–1.79 (m, 2H), 1.59–1.55 (m, 2H), 1.54 (d, *J* = 7.1 Hz, 3H), 1.51 (d, *J* = 7.1 Hz, 3H), 1.50–1.47 (m, 6H), 1.45 (s, 9H), 1.45 (s, 9H), 1.34 (s, 3H), 1.32 (d, *J* = 14.6 Hz, 6H), 1.23–1.13 (m, 2H).

^13^C{^1^H} NMR (151 MHz, CD_3_CN, s 118.26 ppm, 25 °C) *δ* 170.8_8_ (Cq), 170.8_6_ (Cq), 165.5 (Cq), 165.4 (Cq), 155.7_0_ (Cq), 155.6_9_ (Cq), 142.2_2_ (Cq), 142.1_7_ (Cq), 140.0 (Cq), 135.9_3_ (Cq), 135.9_0_ (Cq), 130.5_6_ (CH), 130.5_5_ (CH), 130.1_1_ (Cq), 130.1_0_ (Cq), 129.6 (CH), 128.8_7_ (CH), 128.8_4_ (CH), 127.6_4_ (CH), 127.6_1_ (CH), 126.2 (CH), 126.1 (CH), 124.8_3_ (CH), 124.8_0_ (CH), 123.8_8_ (CH), 123.8_7_ (CH), 122.0 (CH), 121.8 (CH), 119.5 (CH), 119.2 (Cq), 119.1 (Cq), 117.9 (Cq), 117.8 (Cq), 115.6_9_ (Cq), 115.6_7_ (Cq), 114.6 (CH), 114.5 (CH), 99.6 (Cq), 99.5 (Cq), 80.9 (2Cq), 67.2_1_ (CH), 67.1_9_ (CH), 66.7 (CH), 66.5 (CH), 57.3 (CH_3_), 57.2 (CH_3_), 44.0 (CH_2_), 43.9 (CH_2_), 43.5_1_ (CH_2_), 43.4_9_ (CH_2_), 38.8 (CH_2_), 38.6 (CH_2_), 37.1 (CH_2_), 37.0 (CH_2_), 30.5_1_ (CH_3_), 30.4_6_ (CH_3_), 28.3 (CH_3_), 26.6_3_ (CH), 26.5_9_ (CH), 22.4 (CH_3_), 22.2 (CH_3_), 21.4_3_ (CH_3_), 21.4_0_ (CH_3_), 20.3 (CH_3_), 20.1 (CH_3_).

HRMS (ESI-QTOF). Calculated for 641.3591 C_39_H_49_N_2_O_6_ [M + H]^+^; found 641.3598.

### 3.11. T-butyl 2-((4R,6R)-6-(2-(2-bromo-5-isopropyl-3-(2-methoxynaphthalen-1-yl)-4-(phenylcarbamoyl)-1H-pyrrol-1-yl)ethyl)-2,2-dimethyl-1,3-dioxan-4-yl)acetate *(**5e’**)*

In a round-bottom flask equipped with a magnetic stirring bar and under N_2_ atmosphere, the pyrrole **4e’** (0.33 g, 0.52 mmol, 1 eq) was dissolved in anhydrous THF (5 mL, 0.1 M) and cooled to −95 °C. Then, *N*-bromosuccinimide (0.11 mg, 0.57 mmol, 1.1 eq) was added and the solution stirred avoiding light for 2 h, allowing the solution to heat up to −10 °C. Then, the solution was diluted with EtOAc, consequently passed in a silica plug and washed with a saturated solution of NaHCO_3_ (3 × 10 mL). The organic layer was dried over Na_2_SO_4_ and concentrated in vacuo, resulting in the product **5e’** as a pale-yellow solid (0.37 g, yield = 99%). The two diastereoisomers were characterized after CSP-HPLC separation (AD-H, 80:20 Hex/IPA, 20 mL/min, first eluted = 3.7 min, second eluted = 4.9 min).

#### 3.11.1. First Eluted Diastereoisomer (**5e’–1 HPLC**)

^1^H NMR (600 MHz, CD_3_CN, 1.96 ppm, 25 °C) *δ* 8.03–7.98 (m, 1H), 7.91–7.87 (m, 1H), 7.79 (s, 1H), 7.52 (d, *J* = 9.1 Hz, 1H), 7.46–7.41 (m, 1H), 7.41–7.36 (m, 2H), 7.16–7.10 (m, 2H), 7.03–6.99 (m, 2H), 6.93 (td, *J* = 7.4, 1.2 Hz, 1H), 4.35–4.25 (m, 2H), 4.17 (ddd, *J* = 14.7, 10.3, 5.8 Hz, 1H), 4.09 (dddd, *J* = 12.8, 7.9, 4.0, 2.1 Hz, 1H), 3.94 (s, 3H), 3.56 (eptet, *J* = 7.1 Hz, 1H), 2.36 (dd, *J* = 15.0, 4.8 Hz, 1H), 2.28 (dd, *J* = 15.0, 8.1 Hz, 1H), 2.00–1.93 (m, 1H), 1.88 (ddt, *J* = 13.4, 10.3, 6.7 Hz, 1H), 1.64 (dt, *J* = 12.7, 2.5 Hz, 1H), 1.52 (d, *J* = 7.1 Hz, 3H), 1.50 (s, 3H), 1.46 (d, *J* = 7.2 Hz, 3H), 1.45 (s, 9H), 1.35 (s, 3H), 1.24 (dt, *J* = 12.6, 11.4 Hz, 1H).

^13^C{^1^H} NMR (151 MHz, CD_3_CN, s 118.26 ppm, 25 °C) *δ* 170.9 (Cq), 164.7 (Cq), 156.1 (Cq), 143.0 (Cq), 139.6 (Cq), 134.9 (Cq), 131.3 (CH), 130.1 (Cq), 129.6 (CH), 129.1 (CH), 128.1 (CH), 125.6 (CH), 125.0 (CH), 124.2 (CH), 119.6 (CH), 119.0 (Cq), 117.9 (Cq), 116.9 (Cq), 114.8 (CH), 103.5 (Cq), 99.5 (Cq), 80.9 (Cq), 67.4 (CH), 67.1 (CH), 57.5 (CH_3_), 43.5 (CH_2_), 43.5 (CH_2_), 38.3 (CH_2_), 36.8 (CH_2_), 30.4 (CH_3_), 28.3 (CH_3_), 27.6 (CH), 22.4 (CH_3_), 21.4 (CH_3_), 20.2 (CH_3_).

#### 3.11.2. Second Eluted Diastereoisomer (**5e’–2 HPLC**)

^1^H NMR (600 MHz, CD_3_CN, 1.96 ppm, 25 °C) *δ* 8.01 (d, *J* = 9.1 Hz, 1H), 7.88 (dt, *J* = 8.1, 1.0 Hz, 1H), 7.83 (s, 1H), 7.53 (d, *J* = 9.1 Hz, 1H), 7.46–7.35 (m, 3H), 7.16–7.10 (m, 2H), 7.05–6.99 (m, 2H), 6.93 (tt, *J* = 7.3, 1.2 Hz, 1H), 4.35–4.26 (m, 2H), 4.18 (ddd, *J* = 14.7, 9.9, 6.3 Hz, 1H), 4.10–4.06 (m, 1H), 3.95 (s, 3H), 3.55 (eptet, *J* = 7.1 Hz, 1H), 2.36 (dd, *J* = 15.0, 4.8 Hz, 1H), 2.28 (dd, *J* = 15.0, 8.2 Hz, 1H), 1.99 (ddd, *J* = 13.5, 6.5, 3.6 Hz, 1H), 1.86 (dddd, *J* = 13.4, 10.0, 8.4, 4.8 Hz, 1H), 1.63 (dt, *J* = 12.8, 2.5 Hz, 1H), 1.52 (d, *J* = 7.0 Hz, 3H), 1.48 (s, 3H), 1.47 (d, *J* = 7.1 Hz, 3H), 1.45 (s, 9H), 1.34 (d, *J* = 0.8 Hz, 3H), 1.24 (dt, *J* = 12.7, 11.5 Hz, 1H).

^13^C{^1^H} NMR (151 MHz, CD_3_CN, s 118.26 ppm, 25 °C) *δ* 170.9 (Cq), 164.7 (Cq), 156.0 (Cq), 143.0 (Cq), 139.6 (Cq), 134.9 (Cq), 131.3 (CH), 130.1 (Cq), 129.6 (CH), 129.1 (CH), 128.1 (CH), 125.6 (CH), 125.0 (CH), 124.2 (CH), 119.6 (CH), 119.1 (Cq), 116.9 (Cq), 114.8 (CH), 103.6 (Cq), 99.5 (Cq), 80.9 (Cq), 67.2 (CH), 67.1 (CH), 57.5 (CH_3_), 43.5 (CH_2_), 43.4 (CH_2_), 38.4 (CH_2_), 36.9 (CH_2_), 30.4 (CH_3_), 28.3 (CH_3_), 27.6 (CH), 22.5 (CH_3_), 21.3 (CH_3_), 20.2 (CH_3_).

HRMS (ESI-QTOF). Calculated for 719.2696 C_39_H_48_BrN_2_O_6_ [M + H]^+^; found 719.2690.

### 3.12. T-butyl 2-((4R,6R)-6-(2-(2-(4-fluorophenyl)-5-isopropyl-3-(2-methoxynaphthalen-1-yl)-4-(phenylcarbamoyl)-1H-pyrrol-1-yl)ethyl)-2,2-dimethyl-1,3-dioxan-4-yl)acetate *(**1e’**)*

In a round-bottom flask equipped with a magnetic stirring bar and under N_2_ atmosphere, the pyrrole **5e’** (0.58 g, 0.8 mmol, 1 eq) and (4-fluorophenyl)boronic acid (168.8 mg, 1.2 mmol, 1.5 eq) were dissolved in a 19/1 v/v mixture of toluene and CPME (8 mL, 0.1 M). Therefore, a 2 M solution of K_2_CO_3_ (1 mL, 2.5 eq) was added. The solution was degassed with cycles of vacuum/N_2_ in an ultrasonic bath, then a catalytic amount of Pd(PPh_3_)_4_ was added. The resulting mixture was refluxed overnight, avoiding light. The solution was passed through a Celite plug to separate the catalyst, and then a quick silica chromatography column was performed (eluent 65:15:20 mixture of Hex/DCM/EtOAc). The resulting mixture of **1e’** and **4e’** (483 mg, ratio 76:24 of **1e’**/**4e’** by ^1^H-NMR) was finally separated into semi-preparative CSP-HPLC. In this way, it was also possible to separate the diastereoisomers of **1e’** (AD-H, 20 mL/min, 90:10 Hex/IPA). The first diastereoisomer eluted at 3.6 min, collected 131.8 mg, yield = 22%; the second diastereoisomer eluted at 5.7 min, collected 130.4 mg, yield = 22%; pyrrole **4e’** eluted at 6.8 min.

#### 3.12.1. First Eluted Diastereoisomer (**1e’–1 HPLC**)

^1^H NMR (600 MHz, CD_3_CN, 1.96 ppm, 25 °C) *δ* 7.86 (d, *J* = 9.0 Hz, 1H), 7.77 (s, 1H), 7.75 (d, *J* = 8.2 Hz, 1H), 7.44 (d, *J* = 9.1 Hz, 1H), 7.41 (dp, *J* = 8.6, 0.7 Hz, 1H), 7.35 (ddd, *J* = 8.5, 6.7, 1.3 Hz, 1H), 7.28 (ddd, *J* = 8.0, 6.7, 1.3 Hz, 1H), 7.24–7.18 (m, 2H), 7.14–7.08 (m, 2H), 7.01–6.95 (m, 2H), 6.94–6.86 (m, 3H), 4.23–4.15 (m, 1H), 4.10 (ddd, *J* = 14.6, 10.0, 5.9 Hz, 1H), 4.04–3.93 (m, 4H), 3.79 (dddd, *J* = 11.7, 7.2, 4.9, 2.5 Hz, 1H), 3.57 (eptet, *J* = 7.0 Hz, 1H), 2.31–2.17 (m, 2H), 1.79–1.67 (m, 2H), 1.60 (d, *J* = 7.0 Hz, 3H), 1.50 (d, *J* = 7.1 Hz, 3H), 1.43 (s, 8H), 1.40–1.32 (m, 4H), 1.31 (d, *J* = 15.6 Hz, 1H), 1.23 (d, *J* = 0.8 Hz, 3H), 1.00 (dt, *J* = 12.8, 11.5 Hz, 1H).

^13^C{^1^H} NMR (151 MHz, CD_3_CN, s 118.26 ppm, 25 °C) *δ* 170.8 (Cq), 165.5 (Cq), 163.0 (d, *J* = 245.2 Hz, Cq), 156.3 (Cq), 142.4 (Cq), 139.8 (Cq), 135.5 (Cq), 133.3 (CH), 133.2 (CH), 130.8 (Cq), 130.7 (CH), 129.8 (Cq), 129.6 (CH), 128.8 (CH), 127.7 (CH), 125.8 (CH), 124.7 (CH), 124.0 (CH), 119.4 (CH), 118.9 (Cq), 117.9 (Cq), 115.8 (CH), 115.6 (CH), 115.3 (Cq), 114.1 (CH), 99.3 (Cq), 80.9 (Cq), 67.3 (CH), 67.0 (CH), 57.1 (CH_3_), 43.4 (CH_2_), 41.8 (CH_2_), 38.8 (CH_2_), 36.6 (CH_2_), 30.3 (CH_3_), 28.2 (CH_3_), 27.0 (CH), 22.7 (CH_3_), 21.1 (CH_3_), 20.0 (CH_3_). ^19^F NMR (376 MHz, CD_3_CN) *δ*-115.90 (ddd, *J* = 14.6, 9.2, 5.4 Hz).

#### 3.12.2. Second Eluted Diastereoisomer (**1e’–2 HPLC**)

^1^H NMR (600 MHz, CD_3_CN, 1.96 ppm, 25 °C) *δ* 7.86 (d, *J* = 9.1 Hz, 1H), 7.80 (s, 1H), 7.74 (d, *J* = 8.3 Hz, 1H), 7.45 (d, *J* = 9.1 Hz, 1H), 7.41 (dd, *J* = 8.5, 1.1 Hz, 1H), 7.34 (ddd, *J* = 8.5, 6.7, 1.3 Hz, 1H), 7.27 (ddd, *J* = 8.1, 6.7, 1.3 Hz, 1H), 7.23–7.17 (m, 2H), 7.15–7.08 (m, 2H), 7.02–6.96 (m, 2H), 6.94–6.86 (m, 3H), 4.21–4.10 (m, 2H), 4.00 (s, 3H), 3.99–3.92 (m, 1H), 3.80 (dddd, *J* = 11.6, 7.2, 4.5, 2.5 Hz, 1H), 3.56 (eptet, *J* = 7.0 Hz, 1H), 2.28–2.17 (m, 2H), 1.75 (dp, *J* = 15.0, 5.1 Hz, 1H), 1.66 (dddd, *J* = 13.0, 10.2, 7.5, 5.3 Hz, 1H), 1.59 (d, *J* = 7.0 Hz, 3H), 1.51 (d, *J* = 7.1 Hz, 3H), 1.43 (s, 8H), 1.37 (s, 3H), 1.35–1.28 (m, 2H), 1.21 (s, 3H), 0.98 (dt, *J* = 12.7, 11.5 Hz, 1H).

^13^C{^1^H} NMR (151 MHz, CD_3_CN, s 118.26 ppm, 25 °C) *δ* 170.8 (Cq), 165.5 (Cq), 163.0 (d, *J* = 245.3 Hz, Cq), 156.3 (Cq), 142.4 (Cq), 139.8 (Cq), 135.5 (Cq), 133.2 (CH), 133.2 (CH), 130.8 (Cq), 130.7 (CH), 129.8 (Cq), 129.8 (Cq), 129.6 (CH), 128.8 (CH), 127.7 (CH), 125.8 (CH), 124.7 (CH), 124.0 (CH), 119.4 (CH), 118.9 (Cq), 118.0 (Cq), 115.8 (CH), 115.6 (CH), 115.3 (Cq), 114.0 (CH), 99.3 (Cq), 80.9 (Cq), 67.2 (CH), 67.0 (CH), 57.1 (CH_3_), 43.4 (CH_2_), 41.6 (CH_2_), 38.7 (CH_2_), 36.5 (CH_2_), 30.3 (CH_3_), 28.2 (CH_3_), 27.0 (CH), 22.9 (CH_3_), 21.0 (CH_3_), 20.0 (CH_3_).

^19^F NMR (376 MHz, CD_3_CN) *δ*-115.90 (ddd, *J* = 14.5, 9.0, 5.5 Hz).

HRMS (ESI-QTOF). Calculated for 735.3809 C_45_H_52_FN_2_O_6_ [M + H]^+^; found 735.3803.

### 3.13. T-butyl 2-((4R,6R)-6-(2-(4-(2-(benzyloxy)naphthalen-1-yl)-2-isopropyl-3-(phenylcarbamoyl)-1H-pyrrol-1-yl)ethyl)-2,2-dimethyl-1,3-dioxan-4-yl)acetate *(**4h’**)*

In a test tube equipped with a magnetic stirring bar, the enamine solution **2′** (0.880 g, 1.9 mmol, 1 eq) in ethanol (38 mL 0.05 M), (*E*)-2-benzyloxy-1-(2-nitrovinyl)naphthalene **3h** (0.700 g, 2.3 mmol 1.2 eq) and ZrCp_2_Cl_2_ (111 mg, 0.2 eq) were added. The vessel was sealed and the solution was heated to 95 °C for 3 h, then the mixture was filtered to separate the excess of **3h**. The filtrate was concentrated in vacuo. The resulting crude solid was treated over a chromatography column with a mixture of Hex/DCM/EtOAc 70:20:10 to obtain 0.72 g of product **4h’** (yield = 53%) as a pale-yellow solid.

Analyzed as couple of conformational diastereoisomers: ^1^H NMR (600 MHz, CD_3_CN, 1.96 ppm, 25 °C) *δ* 7.96 (d, *J* = 8.7 Hz, 1H), 7.93 (d, *J* = 9.0 Hz, 1H), 7.89 (dd, *J* = 8.0, 1.5 Hz, 2H), 7.87 (dd, *J* = 8.0, 1.5 Hz, 2H), 7.81 (s, 1H), 7.80 (s, 1H), 7.73 (dt, *J* = 3.9, 1.0 Hz, 1H), 7.71 (dt, *J* = 4.1, 1.0 Hz, 1H), 7.54 (d, *J* = 9.0 Hz, 1H), 7.52 (d, *J* = 9.0 Hz, 1H), 7.47–7.43 (m, 2H), 7.42–7.37 (m, 2H), 7.32–7.30 (m, 3H), 7.30–7.29 (m, 4H), 7.29–7.25 (m, 2H), 7.11–7.04 (m, 4H), 6.94–6.85 (m, 6H), 6.57 (s, 1H), 6.56 (s, 1H), 5.27–5.19 (m, 3H), 5.18 (d, *J* = 316.7 Hz, 3H), 4.30–4.24 (m, 1H), 4.19–4.13 (m, 2H), 4.12–4.06 (m, 2H), 4.00–3.95 (m, 1H), 3.89 (dddd, *J* = 11.2, 8.6, 4.4, 2.4 Hz, 1H), 3.80–3.74 (m, 1H), 3.60–3.52 (m, 2H), 2.33 (dd, *J* = 15.0, 4.6 Hz, 1H), 2.27–2.20 (m, 2H), 2.14 (dd, *J* = 14.9, 8.6 Hz, 1H), 1.99–1.93 (m, 2H), 1.93–1.74 (m, 4H), 1.56 (dt, *J* = 12.7, 2.5 Hz, 1H), 1.55–1.51 (m, 8H), 1.49 (d, *J* = 7.1 Hz, 3H), 1.44 (s, 8H), 1.42 (s, 9H), 1.35–1.29 (m, 7H), 1.28 (d, *J* = 13.3 Hz, 6H), 1.25–1.15 (m, 2H), 1.13–1.05 (m, 2H).

^13^C{^1^H} NMR (151 MHz, CD_3_CN, s 118.26 ppm, 25 °C) *δ* 170.9 (Cq), 170.7 (Cq), 165.4 (Cq), 165.3 (Cq), 155.0 (Cq), 154.9 (Cq), 142.5 (Cq), 142.4 (Cq), 139.9 (Cq), 138.3 (Cq), 138.2 (Cq), 135.8_8_ (Cq), 135.8_7_ (Cq), 130.6_1_ (Cq), 130.5_7_ (CH), 130.5 (CH), 129.5 (CH), 129.4_2_ (CH), 129.3_7_ (CH), 128.9_5_ (CH), 128.8_8_ (CH), 128.8 (CH), 128.7 (CH), 128.4 (CH), 128.0 (CH), 127.8 (CH), 127.7 (CH), 126.3 (CH), 126.1 (CH), 125.3 (CH), 123.8 (CH), 122.1 (CH), 121.9 (CH), 121.2 (Cq), 121.1 (Cq), 119.4_1_ (CH), 119.3_7_ (CH), 117.8_1_ (Cq), 117.7_8_ (Cq), 117.4 (CH), 117.2 (CH), 115.6 (Cq), 115.4 (Cq), 99.5_1_ (Cq), 99.4_8_ (Cq), 80.9 (Cq), 80.8 (Cq), 72.8 (CH_2_), 72.7 (CH_2_), 67.2 (CH), 67.0 (CH), 66.4 (CH), 66.1 (CH), 43.9 (CH_2_), 43.8 (CH_2_), 43.5 (CH_2_), 43.4 (CH_2_), 38.9 (CH_2_), 38.7 (CH_2_), 36.9 (CH_2_), 30.4_1_ (CH_3_), 30.3_8_ (CH_3_), 28.3 (CH_3_), 26.7 (CH), 26.6 (CH), 22.2 (CH_3_), 22.0 (CH_3_), 21.5 (CH_3_), 21.4 (CH_3_), 20.2 (CH_3_), 20.0 (CH_3_).

HRMS (ESI-QTOF). Calculated for 717.3904 C_45_H_53_N_2_O_6_ [M + H]^+^; found 717.3910.

### 3.14. T-butyl-2-((4R,6R)-6-(2-(3-(2-(benzyloxy)naphthalen-1-yl)-2-bromo-5-isopropyl-4-(phenylcarbamoyl)-1H-pyrrol-1-yl)ethyl)-2,2-dimethyl-1,3-dioxan-4-yl)acetate *(**5h’**)*

In a round-bottom flask equipped with a magnetic stirring bar and under N_2_ atmosphere, the pyrrole **4h’** (0.42 g, 0.59 mmol, 1 eq) was dissolved in anhydrous THF (6 mL, 0.1 M) and cooled to −95 °C. Then, *N*-bromosuccinimide (0.11 mg, 0.65 mmol, 1.1 eq) was added and the solution stirred, avoiding light, for 2 hours, allowing the solution to heat up to −60 °C after an hour at –95 °C. Then, the solution was diluted with EtOAc, consequently passed in a silica plug and washed with a saturated solution of NaHCO_3_ (3 × 10 mL). The organic layer was dried over Na_2_SO_4_ and concentrated in vacuo, resulting in the product **5h’** as a pale-yellow solid (0.43 g, yield = 91%). The two diastereoisomers were characterized after CSP-HPLC separation (AD-H, 80:20 Hex/IPA, 20 mL/min, first elution = 3.7 min, second elution = 4.7 min).

#### 3.14.1. First Eluted Diastereoisomer (**5h’–1 HPLC**)

^1^H NMR (600 MHz, CD_3_CN, 1.96 ppm, 25 °C) *δ* 7.99 (d, *J* = 8.7 Hz, 1H), 7.90 (dd, *J* = 7.3, 0.8 Hz, 1H), 7.88 (s, 1H), 7.55 (d, *J* = 9.0 Hz, 1H), 7.50 (ddt, *J* = 8.6, 1.6, 0.8 Hz, 1H), 7.49–7.44 (m, 2H), 7.42 (ddd, *J* = 8.1, 6.6, 1.5 Hz, 1H), 7.32 (d, *J* = 1.0 Hz, 1H), 7.32 (s, 2H), 7.31–7.27 (m, 1H), 7.09 (dd, *J* = 8.6, 7.3 Hz, 2H), 6.94 (dd, *J* = 8.7, 1.2 Hz, 2H), 6.91 (tt, *J* = 7.5, 1.2 Hz, 1H), 5.23 (dd, *J* = 50.9, 11.9 Hz, 2H), 4.29 (ddd, *J* = 15.0, 10.3, 4.9 Hz, 1H), 4.26–4.21 (m, 1H), 4.19 (ddd, *J* = 14.7, 10.0, 6.1 Hz, 1H), 4.07–4.01 (m, 1H), 3.59 (eptet, *J* = 7.1 Hz, 1H), 2.34 (dd, *J* = 15.0, 4.8 Hz, 1H), 2.26 (dd, *J* = 15.0, 8.2 Hz, 1H), 1.93 (td, *J* = 6.3, 3.3 Hz, 1H), 1.83 (dddd, *J* = 13.3, 10.0, 8.2, 4.9 Hz, 1H), 1.58 (dt, *J* = 12.7, 2.5 Hz, 1H), 1.53 (d, *J* = 7.1 Hz, 3H), 1.48 (d, *J* = 7.1 Hz, 3H), 1.46 (s, 3H), 1.45 (s, 7H), 1.34 (d, *J* = 0.8 Hz, 3H), 1.21 (dt, *J* = 12.7, 11.5 Hz, 1H).

^13^C{^1^H} NMR (151 MHz, CD_3_CN, s 118.26 ppm, 25 °C) *δ* 170.9 (Cq), 164.6 (Cq), 155.3 (Cq), 143.3 (Cq), 139.6 (Cq), 138.2 (Cq), 134.9 (Cq), 131.3 (CH), 130.6 (Cq), 129.6 (CH), 129.4 (CH), 129.2 (CH), 128.9 (CH), 128.2_0_ (CH), 128.1_8_ (CH), 125.7 (CH), 125.4 (CH), 124.2 (CH), 119.9 (Cq), 119.5 (CH), 119.0 (Cq), 117.4 (CH), 116.9 (Cq), 103.5 (Cq), 99.5 (Cq), 80.9 (Cq), 72.8 (CH_2_), 67.2 (CH), 67.0 (CH), 43.5 (CH_2_), 43.4 (CH_2_), 38.6 (CH_2_), 36.9 (CH_2_), 30.4 (CH_3_), 28.3 (CH_3_), 27.6 (CH), 22.2 (CH_3_), 21.4 (CH_3_), 20.2 (CH_3_).

#### 3.14.2. Second Eluted Diastereoisomer (**5h’–2 HPLC**)

^1^H NMR (600 MHz, CD_3_CN, 1.96 ppm, 25 °C) *δ* 7.98 (d, *J* = 9.0 Hz, 1H), 7.92 (s, 1H), 7.90 (dt, *J* = 8.0, 0.9 Hz, 1H), 7.55 (d, *J* = 9.0 Hz, 1H), 7.50 (ddt, *J* = 8.6, 1.5, 0.8 Hz, 1H), 7.48–7.44 (m, 1H), 7.42 (ddd, *J* = 8.1, 6.6, 1.5 Hz, 1H), 7.32 (d, *J* = 1.0 Hz, 1H), 7.32 (s, 2H), 7.31–7.27 (m, 1H), 7.10 (dd, *J* = 8.6, 7.3 Hz, 2H), 6.99–6.94 (m, 2H), 6.91 (tt, *J* = 7.3, 1.2 Hz, 1H), 5.24 (dd, *J* = 50.7, 11.9 Hz, 2H), 4.35–4.26 (m, 2H), 4.19 (ddd, *J* = 14.7, 9.9, 6.4 Hz, 1H), 4.11–4.03 (m, 1H), 3.58 (eptet, *J* = 7.1 Hz, 1H), 2.36 (dd, *J* = 15.0, 4.8 Hz, 1H), 2.28 (dd, *J* = 15.0, 8.1 Hz, 1H), 1.95–1.91 (m, 1H), 1.86 (dddd, *J* = 13.3, 9.9, 8.3, 4.8 Hz, 1H), 1.61 (dt, *J* = 12.7, 2.5 Hz, 1H), 1.52 (d, *J* = 7.1 Hz, 3H), 1.48 (s, 3H), 1.48 (d, *J* = 7.2 Hz, 3H), 1.45 (s, 7H), 1.34 (d, *J* = 0.8 Hz, 3H), 1.23 (dt, *J* = 12.8, 11.5 Hz, 1H).

^13^C{^1^H} NMR (151 MHz, CD_3_CN, s 118.26 ppm, 25 °C) *δ* 170.9 (Cq), 164.6 (Cq), 155.2 (Cq), 143.3 (Cq), 139.6 (Cq), 138.1 (Cq), 134.9 (Cq), 131.3 (CH), 130.6 (Cq), 129.6 (CH), 129.4 (CH), 129.2 (CH), 128.9 (CH), 128.2_5_ (CH), 128.1_6_ (CH), 125.7 (CH), 125.4 (CH), 124.2 (CH), 119.9 (Cq), 119.5 (CH), 119.0 (Cq), 117.4 (CH), 116.9 (Cq), 103.5 (Cq), 99.5 (Cq), 81.0 (Cq), 72.8 (CH_2_), 67.1 (CH), 67.1 (CH), 43.5 (CH_2_), 43.4 (CH_2_), 38.5 (CH_2_), 36.9 (CH_2_), 30.4 (CH_3_), 28.3 (CH_3_), 27.6 (CH), 22.4 (CH_3_), 21.3 (CH_3_), 20.2 (CH_3_).

HRMS (ESI-QTOF). Calculated for 833.2568 C_45_H_51_BrKN_2_O_6_ [M + K]^+^; found 833.2580.

### 3.15. T-butyl 2-((4R,6R)-6-(2-(3-(2-(benzyloxy)naphthalen-1-yl)-2-(4-fluorophenyl)-5-isopropyl-4-(phenylcarbamoyl)-1H-pyrrol-1-yl)ethyl)-2,2-dimethyl-1,3-dioxan-4-yl)acetate *(**1h’**)*

In a round-bottom flask equipped with a magnetic stirring bar and under N_2_ atmosphere, the pyrrole **5h’** (0.41 g, 0.5 mmol, 1 eq) and (4-fluorophenyl)boronic acid (107 mg, 0.8 mmol, 1.5 eq) were dissolved in a 19/1 *v*/*v* mixture of toluene and CPME (5 mL, 0.1 M). Therefore, a 2 M solution of K_2_CO_3_ (0.7 mL, 2.5 eq) was added. The solution was degassed with cycles of vacuum/N_2_ in an ultrasonic bath, then a catalytic amount of Pd(PPh_3_)_4_ was added. The resulting mixture was refluxed overnight, avoiding light. The solution was passed through a Celite plug to separate the catalyst, then a quick silica chromatography column was performed (eluent 70:20:10 mixture of Hex/DCM/EtOAc). The resulting mixture of **1h’** and **4h’** (334.4 mg, ratio 74:26 **1h’**/**4h’**, by ^1^H-NMR) was finally separated into semi-preparative CSP-HPLC. In this way, it was also possible to separate the diastereoisomers of **1h’** (AD-H, 20 mL/min, 90:10 Hex/IPA). The first diastereoisomer eluted at 5.4 min, collected 142.1 mg, yield = 34%; the second diastereoisomer eluted at 9.0 min, collected 135.6 mg, yield = 33%; pyrrole **4h’** eluted at 12.0 min.

#### 3.15.1. First Eluted Diastereoisomer (**1h’–1 HPLC**)

^1^H NMR (600 MHz, CD_3_CN, 1.96 ppm, 25 °C) *δ* 7.85 (d, *J* = 8.7 Hz, 1H), 7.81–7.75 (m, 2H), 7.57 (dq, *J* = 8.5, 0.9 Hz, 1H), 7.46 (d, *J* = 9.1 Hz, 1H), 7.40 (ddd, *J* = 8.4, 6.8, 1.3 Hz, 1H), 7.33 (d, *J* = 1.2 Hz, 2H), 7.33 (s, 2H), 7.32–7.28 (m, 2H), 7.17 (dd, *J* = 8.5, 5.7 Hz, 2H), 7.08 (dd, *J* = 8.7, 7.2 Hz, 2H), 6.90 (ddt, *J* = 7.3, 2.7, 1.2 Hz, 3H), 6.87 (t, *J* = 8.9 Hz, 3H), 5.29 (dd, *J* = 50.8, 12.3 Hz, 2H), 4.19–4.10 (m, 2H), 4.00 (ddd, *J* = 14.6, 10.4, 5.7 Hz, 1H), 3.76 (dddd, *J* = 11.7, 7.3, 4.3, 2.5 Hz, 1H), 3.63 (eptet, *J* = 7.1 Hz, 1H), 2.25 (dd, *J* = 15.0, 4.9 Hz, 1H), 2.22–2.17 (m, 1H), 1.74 (dddd, *J* = 12.9, 10.5, 7.7, 5.1 Hz, 1H), 1.67 (dddd, *J* = 13.4, 10.2, 5.7, 4.4 Hz, 1H), 1.60 (d, *J* = 7.1 Hz, 3H), 1.54 (d, *J* = 7.0 Hz, 3H), 1.43 (s, 8H), 1.35 (s, 3H), 1.33 (dt, *J* = 12.8, 2.5 Hz, 1H), 1.24 (s, 3H), 0.98 (dt, *J* = 12.8, 11.5 Hz, 1H).

^13^C{^1^H} NMR (151 MHz, CD_3_CN, s 118.26 ppm, 25 °C) *δ* 170.8 (Cq), 165.5 (Cq), 162.9 (d, *J* = 245.2 Hz, Cq), 155.5 (Cq), 142.8 (Cq), 139.8 (Cq), 138.4 (Cq), 135.6 (Cq), 133.3 (CH), 133.2 (CH), 130.7 (Cq), 130.7 (CH), 130.1 (Cq), 129.8 (Cq), 129.8 (Cq), 129.6 (CH), 129.5 (CH), 129.0 (CH), 128.8 (CH), 127.9 (CH), 127.9 (CH), 125.9 (CH), 125.1 (CH), 123.9 (CH), 120.2 (Cq), 119.3 (CH), 117.9 (Cq), 116.1 (CH), 115.8 (CH), 115.7 (CH), 115.4 (Cq), 99.3 (Cq), 80.9 (Cq), 71.7 (CH_2_), 67.2 (CH), 67.0 (CH), 43.4 (CH_2_), 41.8 (CH_2_), 39.0 (CH_2_), 36.6 (CH_2_), 30.4 (CH_3_), 28.3 (CH_3_), 27.1 (CH), 22.4 (CH_3_), 21.5 (CH_3_), 20.1 (CH_3_).

^19^F NMR (376 MHz, CD_3_CN) *δ*-115.75 (ddd, *J* = 14.4, 9.1, 5.4 Hz).

#### 3.15.2. Second Eluted Diastereoisomer (**1h’–2 HPLC**)

^1^H NMR (600 MHz, CD_3_CN, 1.96 ppm, 25 °C) *δ* 7.85 (d, *J* = 8.7 Hz, 1H), 7.82 (s, 1H), 7.78 (dt, *J* = 8.2, 0.8, 0.7 Hz, 1H), 7.56 (dq, *J* = 8.7, 0.9 Hz, 1H), 7.47 (d, *J* = 9.1 Hz, 1H), 7.39 (ddd, *J* = 8.4, 6.7, 1.3 Hz, 1H), 7.34 (d, *J* = 1.1 Hz, 1H), 7.33 (s, 2H), 7.33–7.28 (m, 2H), 7.20–7.13 (m, 2H), 7.10–7.06 (m, 2H), 6.92–6.89 (m, 3H), 6.89–6.84 (m, 2H), 5.30 (dd, *J* = 58.3, 12.1 Hz, 2H), 4.15 (dddd, *J* = 15.7, 14.2, 7.6, 3.7 Hz, 2H), 3.98 (ddd, *J* = 14.7, 10.3, 5.7 Hz, 1H), 3.78 (dddd, *J* = 11.7, 7.2, 4.4, 2.5 Hz, 1H), 3.60 (eptet, *J* = 7.1 Hz, 1H), 2.25 (dd, *J* = 15.0, 4.8 Hz, 1H), 2.19 (dd, *J* = 15.0, 8.1 Hz, 1H), 1.74 (dddd, *J* = 13.4, 10.3, 5.7, 4.4 Hz, 1H), 1.63 (ddd, *J* = 10.3, 7.8, 5.2 Hz, 1H), 1.59 (d, *J* = 7.0 Hz, 3H), 1.53 (d, *J* = 7.0 Hz, 3H), 1.43 (s, 8H), 1.35–1.28 (m, 4H), 1.19 (s, 3H), 0.96 (dt, *J* = 12.7, 11.4 Hz, 1H).

^13^C{^1^H} NMR (151 MHz, CD_3_CN, s 118.26 ppm, 25 °C) *δ* 170.8 (Cq), 165.5 (Cq), 162.95 (d, *J* = 245.3 Hz, Cq), 155.4 (Cq), 142.7 (Cq), 139.8 (Cq), 138.4 (Cq), 135.6 (Cq), 133.3 (CH), 133.2 (CH), 130.6 (Cq), 130.6 (CH), 130.1 (Cq), 129.8 (Cq), 129.8 (Cq), 129.6 (CH), 129.5 (CH), 128.9 (CH), 128.8 (CH), 127.9 (CH), 127.9 (CH), 126.0 (CH), 125.1 (CH), 123.9 (CH), 120.2 (Cq), 119.3 (CH), 117.9 (Cq), 116.1 (CH), 115.8 (CH), 115.7 (CH), 115.4 (Cq), 99.3 (Cq), 80.9 (Cq), 71.7 (CH_2_), 67.1 (CH), 67.0 (CH), 43.4 (CH_2_), 41.6 (CH_2_), 38.9 (CH_2_), 36.5 (CH_2_), 30.3 (CH_3_), 28.2 (CH_3_), 27.1 (CH), 22.5 (CH_3_), 21.3 (CH_3_), 20.0 (CH_3_).

^19^F NMR (376 MHz, CD_3_CN) *δ*-115.83 (ddd, *J* = 14.6, 9.1, 5.4 Hz).

HRMS (ESI-QTOF). Calculated for 849.3681 C_51_H_55_FKN_2_O_6_ [M + K]^+^; found 849.3675.

### 3.16. T-butyl 2-((4R,6R)-6-(2-(2-(4-fluorophenyl)-3-(2-hydroxynaphthalen-1-yl)-5-isopropyl-4-(phenylcarbamoyl)-1H-pyrrol-1-yl)ethyl)-2,2-dimethyl-1,3-dioxan-4-yl)acetate *(**1g’**)*

In a test tube equipped with a magnetic stir bar and with an Argon flow, the corresponding pyrrole **1h’** (100 mg, 0.123 mmol, 1 eq), Pd(OH)_2_ on carbon (20 wt.%, 0.2 eq, 0.0246 mmol, 17 mg) were dissolved on MeOH (3 mL, 0.04 M). Et_3_SiH (1.23 mmol, 200 μL,10 eq) was dropped and the reaction was stirred for 2 h at room temperature. The solution was filtrated through a Celite plug with EtOAc, then concentrated in vacuo. The product was obtained as a white solid without further purification (84.3 mg, 0.117 mmol, 95% yield).

#### 3.16.1. First Eluted Diastereoisomer (**1g’–1 HPLC**)

^1^H NMR (600 MHz, CDCl_3_, 7.26 ppm, TMS, 25 °C) *δ* 7.79–7.76 (m, 1H), 7.74 (d, *J* = 8.8 Hz, 1H), 7.56 (dd, *J* = 8.4, 1.1 Hz, 1H), 7.43 (ddd, *J* = 8.2, 6.8, 1.3 Hz, 1H), 7.32 (ddd, *J* = 8.1, 6.8, 1.2 Hz, 1H), 7.20 (s, 1H), 7.13 (d, *J* = 8.9 Hz, 1H), 7.11–7.05 (m, 2H), 7.04–6.98 (m, 2H), 6.87–6.78 (m, 3H), 6.68–6.62 (m, 2H), 5.61 (s, 1H), 4.26–4.13 (m, 2H), 3.96 (ddd, *J* = 14.4, 10.9, 5.6 Hz, 1H), 3.88 (eptet, *J* = 7.1 Hz, 1H), 3.76–3.67 (m, 1H), 2.39 (dd, *J* = 15.3, 6.9 Hz, 1H), 2.24 (dd, *J* = 15.3, 6.2 Hz, 1H), 1.79–1.66 (m, 2H), 1.59 (dd, *J* = 6.8 Hz, 6H), 1.44 (s, 9H), 1.41–1.33 (m, 4H), 1.31 (s, 3H), 1.06 (dt, *J* = 12.7, 11.5 Hz, 1H).

^13^C{^1^H} NMR (151 MHz, CDCl_3_, 77.16 ppm, TMS, 25 °C) *δ* 170.3 (Cq), 163.4 (Cq), 162.5 (d, *J* = 248.4 Hz), 152.3 (Cq), 145.3 (Cq), 138.3 (Cq), 134.5 (Cq), 131.9 (CH), 131.9 (CH), 131.8 (Cq), 130.5 (CH), 129.0 (Cq), 128.6 (CH), 128.6 (CH), 127.7 (CH), 127.3_5_ (Cq), 127.3_3_ (Cq), 124.2 (CH), 123.8 (CH), 123.3 (CH), 119.3 (CH), 117.4 (CH), 115.7 (Cq), 115.7 (CH), 115.5 (CH), 113.3 (Cq), 111.5 (Cq), 98.9 (Cq), 80.9 (Cq), 66.5 (CH), 66.0 (CH), 42.6 (CH_2_), 41.6 (CH_2_), 38.1 (CH_2_), 36.1 (CH_2_), 30.1 (CH_3_), 28.2 (CH_3_), 26.3 (CH), 21.5 (CH_3_), 21.3 (CH_3_), 19.8 (CH_3_).

^19^F NMR (376 MHz, CDCl_3_) *δ* 112.99 (ddd, *J* = 13.8, 8.8, 5.2 Hz).

#### 3.16.2. Second Eluted Diastereoisomer (**1g’–2 HPLC**)

^1^H NMR (600 MHz, CDCl_3_, 7.26 ppm, TMS, 25 °C) *δ* 7.75 (dd, *J* = 12.0, 8.5 Hz, 2H), 7.51 (d, *J* = 8.4 Hz, 1H), 7.40 (ddd, *J* = 8.4, 6.8, 1.3 Hz, 1H), 7.30 (ddd, *J* = 8.0, 6.7, 1.3 Hz, 1H), 7.22 (s, 1H), 7.15 (d, *J* = 8.9 Hz, 1H), 7.09 (dd, *J* = 8.6, 5.4 Hz, 2H), 7.02 (dd, *J* = 8.5, 7.3 Hz, 2H), 6.88–6.80 (m, 3H), 6.70–6.64 (m, 2H), 5.72 (s, 1H), 4.21 (dd, *J* = 14.8, 7.7 Hz, 1H), 4.16 (dtd, *J* = 13.4, 6.6, 5.4, 2.6 Hz, 1H), 3.98 (dt, *J* = 14.4, 8.0 Hz, 1H), 3.86 (h, *J* = 7.2 Hz, 1H), 3.72 (dtd, *J* = 11.7, 5.8, 2.3 Hz, 1H), 2.39 (dd, *J* = 15.3, 7.0 Hz, 1H), 2.24 (dd, *J* = 15.3, 6.2 Hz, 1H), 1.78–1.68 (m, 2H), 1.59 (dd, *J* = 7.2, 1.3 Hz, 6H), 1.43 (s, 9H), 1.38 (s, 3H), 1.35 (dt, *J* = 12.7, 2.5 Hz, 1H), 1.30 (s, 3H), 1.07 (q, *J* = 11.8 Hz, 1H).

^13^C{^1^H} NMR (151 MHz, CDCl_3_, 77.16 ppm, TMS, 25 °C) *δ* 170.3 (Cq), 163.5 (Cq), 162.2 (d, *J* = 248.3 Hz), 152.5 (Cq), 145.1 (Cq), 138.4 (Cq), 134.4 (Cq), 131.9 (CH), 131.9 (CH), 131.9 (Cq), 130.5 (CH), 129.0 (Cq), 128.6 (CH), 128.5 (CH), 127.6 (CH), 127.4 (Cq), 127.4 (Cq), 124.2 (CH), 123.8 (CH), 123.4 (CH), 119.3 (CH), 117.4 (CH), 115.7 (Cq), 115.6 (CH), 115.5 (CH), 113.4 (Cq), 111.5 (Cq), 98.9 (Cq), 80.9 (Cq), 66.5 (CH), 66.0 (CH), 42.6 (CH_2_), 41.6 (CH_2_), 38.0 (CH_2_), 36.1 (CH_2_), 30.0 (CH_3_), 28.2 (CH_2_), 26.3 (CH), 21.6 (CH_2_), 21.2 (CH_2_), 19.8 (CH_2_).

^19^F NMR (376 MHz, CDCl_3_) *δ*-113.05 (td, *J* = 8.5, 4.3 Hz).

HRMS (ESI-QTOF). Calculated for 721.3653 C_44_H_50_FN_2_O_6_ [M + H]^+^; found 721.3658.

### 3.17. General Procedure for Lactone-Atropostatins *(**7h**)* and *(**7g**)*

#### 3.17.1. 4-(2-(Benzyloxy)naphthalen-1-yl)-5-(4-fluorophenyl)-1-(2-((2*R*,4*R*)-4-hydroxy-6-oxotetrahydro-2H-pyran-2-yl)ethyl)-2-isopropyl-*N*-phenyl-1*H*-pyrrole-3-carboxamide (**7h**)

In a round-bottom flask equipped with a magnetic stirring bar, the pyrrole mixture **1h’** and **4h’** (100 mg, ratio 74:26 **1h’**/**4h’**, 0.0912 mmol **1h’**) was dissolved in a mixture of THF (5 mL). Then, a solution of HCl 5M (400 µL) was added and the mixture stirred for 2 h at +60 °C. The reaction was quenched with saturated solution of NaHCO_3_ at room temperature and the solvents removed by vacuo. The resulting solid was dissolved in DCM (10 mL) and washed with water (2 × 10 mL). The organic layer was dried over Na_2_SO_4_ and concentrated in vacuo, then the product was purified by silica chromatography column (Hex/EtOAc, from 3:1 to 1:1), obtaining 40 mg (0.0575 mmol, Y = 63%) of **7h**. The resulting mixture of **7h** was separated with semi-preparative AD-H column, 20 mL/min, 80:20 Hex/IPA. The first diastereoisomer ***M*-7h** eluted at 5.2 min, collected 18 mg (0.0258 mmol), yield = 28.3%; the second diastereoisomer eluted ***P*-7h** at 7.5 min, collected 17.5 mg (0.0251 mmol), yield = 27.5%.

##### First Eluted Diastereoisomer (**M-7h**)

^1^H NMR (600 MHz, CD_3_CN, 1.96 ppm, 25 °C) δ 7.85 (1H, d, *J* = 9.08 Hz) 7.81 (1H, s) 7.78 (1H, d, *J* = 7.99 Hz) 7.59 (1H, d, *J* = 7.99 Hz) 7.45 (1H, d, *J* = 9.08 Hz) 7.42 (1H, ddd, *J* = 8.36, 6.90, 1.09 Hz) 7.28–7.36 (6H, m) 7.16 (2H, t, *J* = 6.74 Hz) 7.07–7.10 (2H, m) 6.85–6.92 (5 H, m) 5.33 (1H, d, *J* = 12.35 Hz) 5.23 (1H, d, *J* = 12.35 Hz) 4.55 (1H, br dd, *J* = 7.63, 3.63 Hz) 4.16–4.22 (2H, m) 4.08–4.14 (1H, m) 3.57–3.64 (1H, m) 3.18–3.35 (1H, m) 2.58 (1H, dd, *J* = 17.44, 4.72 Hz) 2.38–2.43 (1H, m) 1.85–1.94 (2H, m) 1.68–1.75 (1H, m) 1.59–1.64 (4H, m) 1.54 (3H, d, *J* = 6.90 Hz).

^13^C{^1^H} NMR (151 MHz, CD_3_CN, 118.26 ppm, TMS, 25 °C) δ 170.5 (Cq), 165.5 (Cq), 163.0 (Cq, d, *J* = 246.32 Hz), 155.3 (Cq), 142.7 (Cq), 139.8 (Cq), 138.4 (Cq), 135.7 (Cq), 133.3 (CH), 133.2 (CH), 130.8 (Cq), 130.6 (CH), 130.1 (Cq), 129.6 (CH), 129.5 (CH), 129.0 (CH), 128.8 (CH), 127.9 (CH), 127.9 (CH), 126.0 (CH), 125.1 (CH), 124.0 (CH), 120.1 (Cq), 119.4 (Cq), 116.1 (CH), 115.9 (CH), 115.7 (CH), 115.6 (Cq), 74.1 (CH), 71.7 (CH_2_), 62.9 (CH), 41.8 (CH_2_), 39.1 (CH_2_), 37.9 (CH_2_), 35.9 (CH_2_), 27.1 (CH), 22.5 (CH_3_), 21.5 (CH3).

^19^F NMR (376 MHz, CD_3_CN) δ -115.62 (td, *J* = 9.1, 4.6 Hz).

##### Second Eluted Diastereoisomer (**P-7h**)

^1^H NMR (600 MHz, CD3CN, 1.96 ppm, 25 °C) δ 7.85 (1H, d, *J* = 8.72 Hz) 7.76–7.82 (2H, m) 7.56 (1H, d, *J* = 7.99 Hz) 7.47 (1H, d, *J* = 9.08 Hz) 7.40 (1H, ddd, *J* = 8.36, 6.90, 1.45 Hz) 7.30–7.35 (6H, m) 7.18 (2H, t, *J* = 6.63 Hz) 7.08 (2H, t, *J* = 7.54 Hz) 6.85–6.91 (5H, m) 5.35 (1H, d, *J* = 12.35 Hz) 5.27 (1H, d, *J* = 12.35 Hz) 4.47–4.55 (1H, m) 4.09–4.22 (3H, m) 3.60 (1H, t, *J* = 7.08 Hz) 3.05–3.35 (1H, m) 2.59 (1H, dd, *J* = 17.44, 4.72 Hz) 2.39–2.43 (1H, m) 1.97–2.00 (1H, m) 1.79–1.86 (1H, m) 1.66–1.72 (1H, m) 1.58–1.63 (4H, m) 1.55 (3H, d, *J* = 7.27 Hz).

^13^C{^1^H} NMR (151 MHz, CD_3_CN, 118.26 ppm, TMS, 25 °C) δ 170.5 (Cq) 165.5 (Cq) 163.0 (Cq, d, *J* = 245.23 Hz) 155.4 (Cq) 142.6 (Cq) 139.8 (Cq) 138.4 (Cq) 135.5 (Cq) 133.3 (CH) 133.2 (CH) 130.8 (Cq) 130.6 (CH) 130.1 (Cq) 129.6 (CH) 129.5 (CH) 128.9 (CH) 128.8 (CH) 128.0 (CH) 127.9 (CH) 126.0 (CH) 125.1 (CH) 124.0 (CH) 120.1 (Cq) 119.4 (CH) 116.1 (CH) 115.9 (CH) 115.7 (CH) 115.6 (Cq) 74.1 (CH) 71.7 (CH_2_) 62.8 (CH) 41.8 (CH_2_) 39.1 (CH_2_) 38.0 (CH_2_) 35.9 (CH_2_) 27.1 (CH) 22.4 (CH_3_) 21.5 (CH_3_).

^19^F NMR (376 MHz, CD_3_CN) δ-115.65 (ddd, *J* = 14.6, 9.2, 5.5 Hz).

HRMS (ESI-QTOF). Calculated for 697.3078 C_44_H_42_FN_2_O_5_ [M + H]^+^; found 697.3071.

#### 3.17.2. 5-(4-Fluorophenyl)-1-(2-((2*R*,4*R*)-4-hydroxy-6-oxotetrahydro-2*H*-pyran-2-yl)ethyl)-4-(2-hydroxynaphthalen-1-yl)-2-isopropyl-*N*-phenyl-1*H*-pyrrole-3-carboxamide (**7g**)

In a test tube equipped with a magnetic stir bar and with an Argon flow, the corresponding pyrrole **7h** (15 mg, 0.0215 mmol, 1 eq), Pd(OH)_2_ on carbon (20 wt.%, 0.2 eq, 0.0043 mmol, 3 mg) were dissolved on MeOH (0.5 mL, 0.04 M). Et_3_SiH (0.215 mmol, 35 μL, 10 eq) was dropped and the reaction was stirred for 2 h at room temperature. The solution was filtrated through a Celite plug with EtOAc, then concentrated in vacuo. The product was obtained as a white solid without further purification (12.6 mg, 0.0208 mmol, 97% yield).

##### (**M-7g**)

^1^H NMR (600 MHz, CD_3_CN, 1.96 ppm, 25 °C) δ 7.77–7.86 (1H, m) 7.74 (2H, d, *J* = 8.72 Hz) 7.46 (1H, d, *J* = 8.00 Hz) 7.37 (1H, ddd, *J* = 8.36, 6.90, 1.09 Hz) 7.24–7.30 (3H, m) 7.18 (1H, d, *J* = 8.74 Hz) 7.11 (2H, t, *J* = 7.53 Hz) 6.98 (2H, d, *J* = 7.94 Hz) 6.90–6.94 (3H, m) 4.53 (1H, qd, *J* = 7.75, 3.63 Hz) 4.10–4.22 (3H, m) 3.60 (1H, quin, *J* = 7.18 Hz) 3.18–3.33 (1H, m) 2.60 (1H, dd, *J* = 17.44, 4.72 Hz) 2.40–2.44 (1H, m) 1.91–1.95 (1H, m) 1.84–1.89 (1H, m) 1.68–1.76 (1H, m) 1.58–1.65 (4H, m) 1.53 (3H, d, *J* = 6.90 Hz).

^13^C{^1^H} NMR (151 MHz, CD_3_CN, 118.26 ppm, TMS, 25 °C) δ170.7 (Cq) 165.5 (Cq) 163.1 (Cq, d, *J* = 245.23 Hz) 153.9 (Cq) 143.1 (Cq) 139.8 (Cq) 135.7 (Cq) 133.3 (Cq) 133.2 (CH) 131.7 (Cq) 130.7 (CH) 129.6 (CH) 129.6 (CH) 129.4 (Cq) 129.4 (Cq) 128.9 (CH) 127.8 (CH) 125.5 (CH) 124.3 (CH) 124.1 (CH) 119.4 (CH) 116.0 (CH) 115.8 (CH) 74.3 (CH) 62.8 (CH) 41.9 (CH_2_) 39.1 (CH_2_) 37.8 (CH_2_) 36.0 (CH_2_) 27.1 (CH) 22.5 (CH_3_) 21.3 (CH_3_).

^19^F NMR (376 MHz, CD_3_CN) δ -115.38–115.52 (m).

##### (**P-7g**)

^1^H NMR (600 MHz, CD_3_CN, 1.96 ppm, 25 °C) δ 7.83 (s, 1H), 7.76–7.71 (m, 2H), 7.48 (dp, *J* = 8.5, 0.7 Hz, 1H), 7.37 (ddd, *J* = 8.3, 6.8, 1.3 Hz, 1H), 7.31–7.23 (m, 3H), 7.18 (d, *J* = 8.9 Hz, 1H), 7.14–7.08 (m, 2H), 7.01–6.95 (m, 2H), 6.95–6.88 (m, 3H), 4.53 (ddt, *J* = 11.6, 7.8, 3.7 Hz, 1H), 4.20–4.16 (m, 1H), 4.11 (ddd, *J* = 15.0, 9.1, 6.3 Hz, 1H), 3.59 (eptet, *J* = 7.0 Hz, 1H), 3.37 (d, *J* = 9.0 Hz, 1H), 2.59 (dd, *J* = 17.5, 4.7 Hz, 1H), 2.42 (ddd, *J* = 17.5, 3.6, 1.8 Hz, 1H), 2.16 (s, 33H), 1.95–1.86 (m, 2H), 1.77–1.70 (m, 1H), 1.63 (ddd, *J* = 14.3, 11.3, 3.1 Hz, 1H), 1.59 (d, *J* = 7.1 Hz, 3H), 1.53 (d, *J* = 7.1 Hz, 3H).

^13^C{^1^H} NMR (151 MHz, CD_3_CN, 118.26 ppm, 25 °C) δ 170.8 (Cq), 165.4 (Cq), 163.8 (Cq, d, *J* = 245.23 Hz) 162.3 (Cq), 153.9 (Cq), 143.2 (Cq), 139.8 (Cq), 135.7 (Cq), 133.3 (Cq), 133.2 (CH), 131.8 (Cq), 130.7 (CH), 129.6 (CH), 129.6 (CH), 129.4 (Cq), 128.9 (CH), 127.8 (CH), 125.5 (CH), 124.3 (CH), 124.0 (CH), 119.4 (CH), 115.9 (CH), 115.8 (CH), 74.3 (CH), 62.8 (CH), 42.0 (CH_2_), 39.1 (CH_2_), 37.7 (CH_2_), 35.9 (CH_2_), 27.1 (CH), 22.5 (CH_3_), 21.2 (CH_3_).

^19^F NMR (376 MHz, CD_3_CN) δ-115.49 (dq, *J* = 14.5, 5.6, 4.6 Hz).

HRMS (ESI-QTOF). Calculated for 607.2608 C_37_H_36_FN_2_O_5_ [M + H]^+^; found 607.2614.

## 4. Conclusions

In summary, in this manuscript we have proposed a novel atropisomeric candidate lactone-atropostatin ***M*-7g** prodrug inhibitor of HMG-CoA reductase. We firstly synthesized and evaluated in silico a library of atorvastatin models with different aryls in position 3 of the pyrrole scaffold in order to obtain stable atropisomers. Then, we determined that *M*-2-OH-naphthyl-atropostatin was the most active inhibitor by means of docking simulation. We were able to obtain the full synthesis of lactone atorvastatin ***M*-7g** with satisfactory overall yields (10%) after six synthetic steps. Hereafter, in vivo screenings could be evaluated with this prodrug.

## Data Availability

Not applicable.

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
