# Peer review of "Atropostatin: Design and Total Synthesis of an Atropisomeric Lactone–Atorvastatin Prodrug"

_molecules, 2023, doi:10.3390/molecules28073176_

Round 1

Reviewer 1 Report

This manuscript is a quit interesting from methodological point of view. Authors made the theoretical calculation to design the molecules with the stable absolute configuration and obtained few of them to carried out next the biological studies in silicon.

The manuscript is prepared well and it should be considered for the publication. Nevertheless, several weaker sides are still present there:

a) The schemes and pictures are made in low quality and should be improved.

b) The assumption that the DFT calculation of racemisation barrier are correct should be proved e.g. by comparison with well known studies published by other research groups (for example 10.5267/j.ccl.2015.6.003 )

c) The docking studies seems to indicates that the affinity of R and S isomers of designed compounds are almost the same in the majority of the presented cases, and this may means that: 1) the calculation were not designed properly, 2) the main assumption that absolute configuration of the compounds influences on their activity is not correct; 3) the racemisation barrier is too low and both R and S isomers adopt the most effective conformation forced by protein. This should be addressed in the manuscript.

d) The asymmetric cross coupling reaction are already studied in detail (for example 10.3390/catal13020353) including the mechanism of such transformations that should be addressed in the manuscript and considered during the synthesis of the new chiral compounds.

e) The determination of absolute configuration is based on comparison of experimental and calculated CD spectra. Nevertheless, the evidence that calculated spectra were performed correctly is not provided. There are a lot of parameters that may influence on the calculated outcome: theoretical mode, basis set, even software used. It is obvious that the calculation will provide a results whatever the input data are correct or false. Authors should prove that the calculation are correct. The bast way is record the CD of compound with very similar structure (for example 10.1016/j.tetasy.2008.10.001) and known configuration or at least calculate it CD using the same calculation settings (for example in manuscript 10.1002/chir.22750 you could find some idea how to do it properly).

f) And last, at least for one compound with the bigger difference of calculated energy of binding to protein in vitro biological test should be presented to verify the postulates of the studies.

Once the Authors fix those methodological gaps the manuscript could be accepted.

Author Response

This manuscript is a quit interesting from methodological point of view. Authors made the theoretical calculation to design the molecules with the stable absolute configuration and obtained few of them to carried out next the biological studies in silicon.

The manuscript is prepared well and it should be considered for the publication. Nevertheless, several weaker sides are still present there:

  1. a) The schemes and pictures are made in low quality and should be improved.

Answer:

The schemes and pictures have been improved.

  1. b) The assumption that the DFT calculation of racemisation barrier are correct should be proved e.g. by comparison with well known studies published by other research groups (for example 10.5267/j.ccl.2015.6.003 )

Answer:

The DFT calculations for racemization barrier were performed in order to forecast which compounds could yield stable atropisomers. The experimental values were determined by kinetic experiments (see SI).  Some additional references have been added. (J. Org. Chem. 2023, 88, 871–881, doi: 10.1021/acs.joc.2c02209; J. Org. Chem. 2019, 84, 12253–12258, doi: 10.1021/acs.joc.9b01550, Current Chemistry Letters 2015, 4, 145-152, doi: 10.5267/j.ccl.2015.6.003).

  1. c) The docking studies seems to indicates that the affinity of R and S isomers of designed compounds are almost the same in the majority of the presented cases, and this may means that: 1) the calculation were not designed properly, 2) the main assumption that absolute configuration of the compounds influences on their activity is not correct; 3) the racemisation barrier is too low and both R and S isomers adopt the most effective conformation forced by protein. This should be addressed in the manuscript.

Answer:

We agree that in some compounds the binding energy of the M/P diastereoisomer is very similar. However, the in-silico screening was performed to identify possible candidates to be synthesized (i.e. IIg in Table 2). In all the tested atropostatins, the affinity was better that the known atorvastatin.

Point 3: the conformational analysis was done on the compound with o-tolyl, which has a small rotational barrier. However, later on in the manuscript more hindered compounds were designed and prepared, and the experimental barriers (experimentally measured see SI and table 1) are bigger than 30 kcal/mol, thus they are fully stable even a +37 °C. Table 1 has been update with the half-life time at +37°C.

  1. d) The asymmetric cross coupling reaction are already studied in detail (for example 10.3390/catal13020353) including the mechanism of such transformations that should be addressed in the manuscript and considered during the synthesis of the new chiral compounds.

Answer:

The last coupling reaction does not involve the forge of any axial chirality. 4-fluorophenylboronic acid is a C2-symmetric compound, and we used fully standard reaction conditions- Thus the suggested manuscript has no reasons to be cited.

  1. e) The determination of absolute configuration is based on comparison of experimental and calculated CD spectra. Nevertheless, the evidence that calculated spectra were performed correctly is not provided. There are a lot of parameters that may influence on the calculated outcome: theoretical mode, basis set, even software used. It is obvious that the calculation will provide a result whatever the input data are correct or false. Authors should prove that the calculation are correct. The best way is record the CD of compound with very similar structure (for example 10.1016/j.tetasy.2008.10.001) and known configuration or at least calculate it CD using the same calculation settings (for example in manuscript 10.1002/chir.22750 you could find some idea how to do it properly).

Answer:

Within our research group we have some skills about the determination of absolute configuration using optical activity coupled with TD-DFT calculations. We updated the reference with:

1)         JOC 2023, 88, 871-881, doi: 10.1021/acs.joc.2c02209;

2)         JOC 2019, 84, 12253-12258, doi: 10.1021/acs.joc.9b01550;

3)         RSC ADVANCES, 2019, 9, 18165 – 18175, doi: 10.1039/C9RA03526E;

4)         CHEMISTRY-A EUROPEAN JOURNAL, 2018, 24, 13306 – 13310, doi: 10.1002/chem.201802303;

5)         CHEMICAL SCIENCE, 2018, 9, pp. 6368 – 6373, doi.org/10.1039/C8SC00913A

6)         JOC, 2017, 82, 6874 – 6885, doi: 10.1021/acs.joc.7b01010

7)         CHIRALITY 2016, 28, 466–474, doi: 10.1002/chir.22600

Every time the simulated AC was compared with anomalous dispersion X-Ray data we have checked that our simulation was correct (see for example RSC Advances 2019, for a very difficult case study). About the suggestion of the referee, we would like to stress that a large part of the section is explained in detail into Materials and Methods and into the Supporting information file.

  1. f) And last, at least for one compound with the bigger difference of calculated energy of binding to protein in vitro biological test should be presented to verify the postulates of the studies.

Once the Authors fix those methodological gaps the manuscript could be accepted.

Answer:

Some contacts with colleagues to design some in-vitro studies are underway. However, the present manuscript paves the in-silico design and synthetic way to these new compounds, and we believe the biological follow-up has to be reported into another manuscript. We prepare the manuscript in order to appear into the special issue called “Synthesis and Application of Atropisomeric Molecules”.

Reviewer 2 Report

The authors present a study of atropisomerism in atorvastatin-like analogues, with prospective design of potential modifications of current clinical lead. QM, NMR, ECD, HPLC and docking studies were applied to evaluate the impact of enabling atropisomerism candidates in these chemical space.

Despite of the amount of information, draft fail to address fundamental questions related to design and scope of new HMGR hits. Additionally, several methodologies are out-dated (B3LYP DFT) or insufficiently explained. These are as follows:

1)    Introduction: What is the desired pharmacological effect with these new hits? Lines 54-55 don´t describe them, helping to understand if they´re searching inhibition, activation, potency or selectivity. Authors should clarify the pharmacological aims of the study, as well as stability expected properties of these analogues.

Line 49: postion 4 is occupied by a pheny-amide, not a carbamate.

Line 177: Figure 2 instead of 3.

2)    DFT calculations: author´s used the out-dated B3LYP DFT functional, as it was demonstrated to posses a high RMSE in extensive energetic and geometric benchmark studies. [1] Authors are strongly encouraged to repeat the overall transition state studies invoking a improved functional, like D3-B3LYP. Is important to clarify that the existence of similar literature publications using the same functional should not be used as justification; otherwise, community will continue growing up the number of articles using a biased theory (B3LYP) and consistently self-supporting each other.

Authors should use Gibbs free energy as thermodynamic descriptors (instead of current electronic energies) when comparing with macromolecular magnitudes obtained from experimental data. Additionally, QM software description is absence.

Are the QM calculations solvent-based or vacuum-based? How they correlate with experimental NMR?

3)    Kinetic studies: author´s don´t explain sufficiently the foundations of the kinetic methodologies. What are the parameters Ca/Caeq? Are they concentrations? It is difficult to interpret results without a clear description of the steps taken to obtain the described correlation models, and hence, experimental Gibbs free energies.

Is it meaningful to compare atropisomeric ratios of interconvertible isomers under different solvent conditions (CSP-HPLC)?

In line with the introduction, authors should clarify their aims in terms of atropisomeric stability. At the light of the results (which should be amended following point 2-3 lines), how valuable are these designs from the MedChem point of view? Results will classify them as Class 2 atropisomers,[2] constituting a serious handicap due to regulatory-based complications. It is understandable these compounds should serve as tool compounds to study a suitable stability

threshold; however, as authors didn´t address proper stability objectives, a proper assessment of their value and prospective modifications is lacking. Rational discussion of these is paramount to build a solid argument.

4)    Molecular docking: a proper description of receptor and ligands preparation is missing (pH, relaxation, etc). Docking methodology is not described as well (AutoDock 4 parameters of Genetic Algorithm search or stochastic one). A docking validation of this methodology should be incorporating, redocking the crystallized PDB ligand and checking the theoretical to experimental RMSE.

When incorporating images in the document, authors are encouraged to maximize the resolution: Figure 2 doesn´t add any value to the manuscript as the quality is very poor.

A convenient description of the binding topologies should be attached as well. Are they any impacts in position 4 amide torsions? The poor-quality Figure 2 image shows a different torsional profile of this group, losing a hydrogen bond from atorvastatin left-image.

AutoDock docking scores are retrieved as Binding energies. In general, docking scores should be considered individually to each ligand (bind-no-bind) before comparing two molecules. The small differences in the scores (less than 0.4 Kcal/mol) together with Class-2 atropisomeric stability should open a discussion around the biological value of activating atropisomerism in these compounds, provided they will racemize and the results of the M/P atropisomers are very close.

Authors are encouraged to perform more accurate theoretical binding measurements via MD simulations and FEP or MMGBSA (or PBSA) protocols.

1.                             Li, A.; Muddana, H.S.; Gilson, M.K. Quantum Mechanical Calculation of Noncovalent Interactions: A Large-Scale Evaluation of PMx, DFT, and SAPT Approaches. Journal of Chemical Theory and Computation 2014, 10, 1563-1575. [10.1021/ct401111c].

2.                             Basilaia, M.; Chen, M.H.; Secka, J.; Gustafson, J.L. Atropisomerism in the Pharmaceutically Relevant Realm. Acc. Chem. Res. 2022, 55, 2904-2919. [10.1021/acs.accounts.2c00500].

Author Response

The authors present a study of atropisomerism in atorvastatin-like analogues, with prospective design of potential modifications of current clinical lead. QM, NMR, ECD,HPLC and docking studies were applied to evaluate the impact of enabling atropisomerism candidates in these chemical space.

Despite of the amount of information, draft fail to address fundamental questions related to design and scope of new HMGR hits. Additionally, several methodologies are  out-dated  (B3LYP DFT) or insufficiently explained.

These are as follows:

1)    Introduction: What is the desired pharmacological effect with these new hits? Lines 54-55 don´t describe them, helping to understand if they´re searching inhibition, activation, potency or selectivity. Authors should clarify the pharmacological aims of the study, as well as stability expected properties of these analogues.

Answer:

The main idea of this research was to enhance the binding constant of known atorvastatine (i.e. inhibition of HMGR enzyme) by adding non-conventional chirality in a molecular branch where conventional chirality was unfeasible. We have modified the introduction to better explain this point.

Line 49: postion 4 is occupied by a pheny-amide, not a carbamate.

Line 177: Figure 2 instead of 3.

Answer:

Line 49 and 177 have been corrected.

2)    DFT calculations: author´s used the out-dated B3LYP DFT functional, as it was demonstrated to possess a high RMSE in extensive energetic and geometric benchmark studies. [1] Authors are strongly encouraged to repeat the overall transition state studies invoking a improved functional, like D3-B3LYP. Is important to clarify that the existence of similar literature publications using the same functional should not be used as justification; otherwise, community will continue growing up the number of articles using a biased theory (B3LYP) and consistently self-supporting each other.

Authors should use Gibbs free energy as thermodynamic descriptors (instead of current electronic energies) when comparing with macromolecular magnitudes obtained from experimental data. Additionally, QM software description is absence.

Are the QM calculations solvent-based or vacuum-based? How they correlate with experimental NMR?

Answer:

We want to thank the review2 for the suggestion to use an improved functional. The values obtained with EmpiricalDispersion=GD3 are closer to the experimental values. Therefore, the table 1 has been corrected using B3LYP as functional, 6-31G(d) as basis set and with EmpiricalDispersion=GD3 as request by reviewer2. Also, the supporting information was updated with these new calculation results.

3)    Kinetic studies: author´s don´t explain sufficiently the foundations of the kinetic methodologies. What are the parameters Ca/Caeq? Are they concentrations? It is difficult to interpret results without a clear description of the steps taken to obtain the described correlation models, and hence, experimental Gibbs free energies.

Answer:

The supporting information was update to better understand the kinetic studies.

Is it meaningful to compare atropisomeric ratios of interconvertible isomers under different solvent conditions (CSP-HPLC)?

Answer:

The atropisomeric ratios could change only with temperature. The life-time was reported in table1. CSP-HPLC was performed at room temperature, therefore atropisomeric ratio doesn’t change.

In line with the introduction, authors should clarify their aims in terms of atropisomeric stability. At the light of the results (which should be amended following point 2-3 lines), how valuable are these designs from the MedChem point of view? Results will classify them as Class 2 atropisomers,[2] constituting a serious handicap due to regulatory-based complications. It is understandable these compounds should serve as tool compounds to study a suitable stability threshold; however, as authors didn´t address proper stability objectives, a proper assessment of their value and prospective modifications is lacking. Rational discussion of these is paramount to build a solid argument.

Answer:

We have updated Table 1 to indicate the half-life times derived by the experimental determination of racemization barriers. It comes out that 75% of the presented compounds belong to class III atropisomers.

The definition of “Class II” statins (see introduction in the original manuscript) is a different definition that is not referred to the conformational stability of the atropisomers, where the “Class X” nomenclature is referred to the measurement of the racemization barrier. Our target was, since the beginning, to prepare fully stable atropisomers, called “Class III” using the classification proposed by LaPlante and others.

The Introduction was updated with the aim to better clarify our target.

4)    Molecular docking: a proper description of receptor and ligands preparation is missing (pH, relaxation, etc). Docking methodology is not described as well (AutoDock 4 parameters of Genetic Algorithm search or stochastic one). A docking validation of this methodology should be incorporating, redocking the crystallized PDB ligand and checking the theoretical to experimental RMSE.

When incorporating images in the document, authors are encouraged to maximize the resolution: Figure 2 doesn´t add any value to the manuscript as the quality is very poor.

A convenient description of the binding topologies should be attached as well. Are they any impacts in position 4 amide torsions? The poor-quality Figure 2 image shows a different torsional profile of this group, losing a hydrogen bond from atorvastatin left-image.

AutoDock docking scores are retrieved as Binding energies. In general, docking scores should be considered individually to each ligand (bind-no-bind) before comparing two molecules. The small differences in the scores (less than 0.4 Kcal/mol) together with Class-2 atropisomeric stability should open a discussion around the biological value of activating atropisomerism in these compounds, provided they will racemize and the results of the M/P atropisomers are very close.

Authors are encouraged to perform more accurate theoretical binding measurements via MD simulations and FEP or MMGBSA (or PBSA) protocols.

  1. Li, A.; Muddana, H.S.; Gilson, M.K. Quantum Mechanical Calculation of Noncovalent Interactions: A Large-Scale Evaluation of PMx, DFT, and SAPT Approaches. Journal of Chemical Theory and Computation 2014, 10, 1563-1575. [10.1021/ct401111c].
  2. Basilaia, M.; Chen, M.H.; Secka, J.; Gustafson, J.L. Atropisomerism in the Pharmaceutically Relevant Realm. Acc. Chem. Res. 2022, 55, 2904-2919. [10.1021/acs.accounts.2c00500].

Answer:

We decided to use the AutoDock 4.2 protocol because some of us was already experienced with this software, and because we needed preliminary results for the evaluation of possible candidates. Supporting file and the main text were updated with more details.

As from the new Table 1 containing the half-life times, many of the reported compounds belong to class 3 atropisomers, not class 2. We also added the suggested literature [4].

Reviewer 3 Report

This manuscript by Mancinelli and coworkers reported the total synthesis of a lacton-atorvastatin prodrug with additional atropisomeric features. In addition, they performed the conformational and experimental studies of model compounds to test the stability of chiral axis. More importantly, docking calculations have been performed to evaluate the constant inhibition of a library of atorvastatins. Due to the importance of atorvastatin derivatives in pharmaceuticals and the unique features of axially chiral compounds, this work is recommended for publication in Molecules after addressing the following issues.

1. In the introduction section, the authors should mention the unique features of axially chiral compounds and the importance of the synthesis of such compounds with the citation of some recent related reviews and publications on this topic, particularly those involving the synthesis of axially chiral heterocyclic compounds. For recent reviews: Acc. Chem. Res. 2022, 55, 2562; Acc. Chem. Res. 2022, 55, 2920; For recent publications: Chin. J. Chem. 2022, 40, 2151; Angew. Chem. Int. Ed. 2023, 62, e202215820; Chem. Synth. 2023, 3, 6; Chem. Synth. 2023, 3, 11.

2. In Scheme 2, the range of the yields of products 4, 5, 1 should be provided below their structures.

3. In the section of “3. Materials and Methods”, “(E)” should be italic, and the exact weight (in mg) of each product should be added behind the yield. In addition, the melting point should be given when the product is solid.

4. In the supplementary materials, in the section of “Scheme of synthesis”, the detailed experimental procedure for each scheme should be given, and the yield of product should be added below the structure of the product.

Author Response

This manuscript by Mancinelli and coworkers reported the total synthesis of a lacton-atorvastatin prodrug with additional atropisomeric features. In addition, they performed the conformational and experimental studies of model compounds to test the stability of chiral axis. More importantly, docking calculations have been performed to evaluate the constant inhibition of a library of atorvastatins. Due to the importance of atorvastatin derivatives in pharmaceuticals and the unique features of axially chiral compounds, this work is recommended for publication in Molecules after addressing the following issues.

  1. In the introduction section, the authors should mention the unique features of axially chiral compounds and the importance of the synthesis of such compounds with the citation of some recent related reviews and publications on this topic, particularly those involving the synthesis of axially chiral heterocyclic compounds. For recent reviews: Acc. Chem. Res. 2022, 55, 2562; Acc. Chem. Res. 2022, 55, 2920; For recent publications: Chin. J. Chem. 2022, 40, 2151; Angew. Chem. Int. Ed. 2023, 62, e202215820; Chem. Synth. 2023, 3, 6; Chem. Synth. 2023, 3, 11.

Answer:

References have been added.

  1. In Scheme 2, the range of the yields of products 4, 5, 1 should be provided below their structures.

The range of the yields have been added in all schemes.

  1. In the section of “3. Materials and Methods”, “(E)” should be italic, and the exact weight (in mg) of each product should be added behind the yield. In addition, the melting point should be given when the product is solid.

The corrections have been done.

  1. In the supplementary materials, in the section of “Scheme of synthesis”, the detailed experimental procedure for each scheme should be given, and the yield of product should be added below the structure of the product.

The details experimental procedure have been added.

Round 2

Reviewer 1 Report

Authors modified some of the highlighted issues, nevertheless in some cases the propositions were not considered. The manuscript is certainly prepared better then in previous version, but still could be improved according the suggestions already provided.

Author Response

"In general Authors provide only cosmetic corrections, and the manuscript is misleading since the only arguments supporting the title, they used is docking studies (which actually do not support the thesis, since in the majority of the cases P and M isomers were calculated as similarly active, and only in few cases by circumstances the activity was not identical). Next or rather before of that Authors make some compounds but did not test them. Methodologically any of assumptions Authors used were supported. So, it is generally having no sense.

But the synthesis is interesting.

Technically manuscript prepared well, it just not contained cohesive data."

Answer:

We understand the issue raised by the reviewer, and we thank them for pointing this out. Therefore, we have changed the title into a more focused topic: “Atropostatin: design and total synthesis of an atropisomeric lacton-atorvastatin prodrug.”

Reviewer 2 Report

Authors should incorporate D3 to all B3LYP text instances, like B3LYP-D3/6-31G.

Author Response

We thank the reviewer. We added D3 to all B3LYP.